# Role Bias in Diffusion Models: Diagnosing and Mitigating through Intermediate Decomposition

**Sina Malakouti**
University of Pittsburgh
Pittsburgh, PA
sem238@pitt.edu

**Adriana Kovashka**
University of Pittsburgh
Pittsburgh, PA
kovashka@cs.pitt.edu

https://sinamalakouti.github.io/ReBind/

## Abstract

Text-to-image (T2I) diffusion models exhibit impressive photorealistic image generation capabilities, yet they struggle in compositional image generation. In this work, we introduce RoleBench, a benchmark focused on evaluating compositional generalization in action-based relations (e.g., "mouse chasing cat"). We show that state-of-the-art T2I models and compositional generation methods consistently default to frequent reversed relations (i.e., "cat chasing mouse"), a phenomenon we call role collapse. Related works attribute this to the model's architectural limitation or underrepresentation in the data. Our key insight reveals that while models fail on rare compositions when their inversions are common, they can successfully generate similar intermediate compositions (e.g., "mouse chasing boy"), suggesting that this limitation is also due to the presence of frequent counterparts rather than just the absence of rare compositions. Motivated by this, we hypothesize that directional decomposition can gradually mitigate role collapse. We test this via ReBind, a lightweight framework that teaches role bindings using carefully selected active/passive intermediate compositions. Experiments suggest that intermediate compositions through simple fine-tuning can significantly reduce role collapse, with humans preferring ReBind more than 78% compared to state-of-the-art methods. Our findings highlight the role of distributional asymmetries in compositional failures and offer a simple, effective path for improving generalization.

## 1 Introduction

Text-to-image (T2I) diffusion models generate high-quality, photorealistic images from text, yet they struggle with spatial understanding, complex descriptions, attribute binding, and compositional generation [Chatterjee et al., 2024, Samuel et al., 2024, Cong et al., 2025, Trusca et al., 2024, Zeng et al., 2023]. Existing methods improve spatial reasoning and attribute binding through attention manipulation [Chefer et al., 2023, Chen et al., 2024], layout conditioning [Li et al., 2023, Feng et al., 2023b], and LLM-based methods [Lian et al., 2024, Yang et al., 2024, Wu et al., 2024a]. However, these approaches are typically evaluated on simple spatial relations (e.g., left/right) and basic attributes (e.g., color and size), leaving **their ability to generalize to rare (unseen) compositions underexplored.**

We explore a more challenging form of compositional generalization: **action-based relations with unusual directions**, i.e., where typical agent-patient roles are reversed (e.g., "*mouse* chasing *cat*"). Prior works have mainly studied static compositions involving rare objects and/or unusual attributes [Samuel et al., 2024, Cong et al., 2025, Park et al., 2025]. In contrast, we focus on dynamic action-based relations that require correct **role binding** between subject (agent) and object

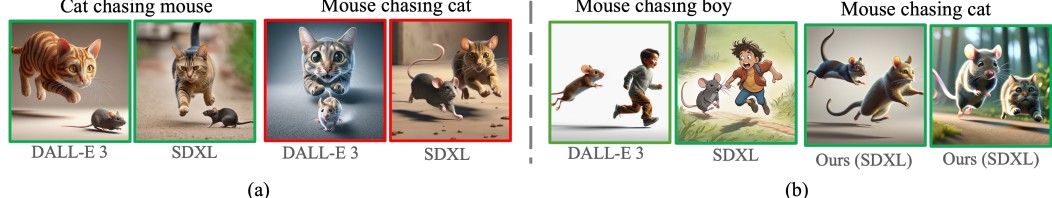

| Cat chasing mouse | Mouse chasing cat | | Mouse chasing boy | Mouse chasing cat |
| DALL-E 3    SDXL | DALL-E 3    SDXL | | DALL-E 3    SDXL | Ours (SDXL)    Ours (SDXL) |
| (a) | | | (b) | |

Figure 1: **Directional bias in action generation.** (a) **Role collapse**: T2I models reliably generate frequent compositions (e.g., "cat chasing mouse") but fail on rare cases (e.g., "mouse chasing cat"), defaulting to the frequent form. (b) Intermediate compositions (e.g., "mouse chasing boy") can be used to enable models to correctly depict rare compositions. Colors: correct / incorrect generation.

(patient) that reflects real-world dynamics. For example, simply placing a mouse behind a cat is insufficient to convey "mouse chasing cat". The model must generate an appropriate spatial configuration (behind), orientation, pose (leaning forward), and facial expression (gaze). To study this phenomenon, we introduce RoleBench, a methodically constructed benchmark designed for evaluating **directional role bias in action-based relations**. RoleBench includes 10 common action verbs, each paired with *frequent* (e.g., "cat chasing mouse") and reversed *rare* counterparts (i.e., "mouse chasing cat"). We construct RoleBench using LLM-estimated semantic plausibility, shown to correlate with frequency in training data [Kauf et al., 2024, Pedinotti et al., 2021], to generate and validate frequent/rare compositions. This design isolates role bias by studying rare but plausible compositions rather than impossible ones (e.g., "chair chasing cat"). We find that T2I models and compositional generation methods **consistently collapse to the frequent composition** (Fig. 1.a). However, they succeed at generating *similarly rare* ones like "mouse chasing boy" (Fig. 1.b). This pattern suggests that compositional failure arises not just from the absence of rare compositions but from the overrepresentation of their frequent counterparts.

Inspired by these findings, we hypothesize that models can *generalize to rare compositions by being exposed to similar yet more learnable compositions*. Our *key insight* is that direct learning of rare compositions is hindered by strong priors in training data, but can be facilitated through intermediate compositions that bridge the gap between frequent and rare patterns. To test this hypothesis, we propose **ReBind**, a simple yet effective compositional framework that decomposes rare directional relations into more plausible compositions while preserving their semantic intent. ReBind uses an LLM to decompose a rare composition (e.g., "mouse chasing cat") into two types of intermediate compositions: (1) *Active intermediates* preserve the subject in an active role (e.g., "mouse chasing boy"). (2) *Passive intermediates* preserve the object in a passive role (e.g., "girl chasing cat"). Fine-tuning on these intermediates reinforces underrepresented role bindings and enables the model to reconstruct the intended rare relation more faithfully. Unlike training-free compositional methods, ReBind achieves this without additional inference-time overhead or architectural modifications.

We evaluate ReBind on RoleBench with VQAScore [Lin et al., 2024], a state-of-the-art compositionality metric. However, such global alignment scores often fail to capture directional role binding. To address this, we adapt VQAScore by adding binary questions that probe fine-grained cues, such as relative position, gaze, and pose (e.g., "Is the mouse *behind* the cat?"). We further introduce a directional bias score, defined as the difference in alignment scores between a prompt and its reversed variant, to quantify the model's tendency to default to frequent directions.

To summarize, our contributions are: (1) We introduce RoleBench, a controlled benchmark targeting directional role bias in action-based relations. (2) We identify role collapse, a failure mode where overrepresentation of frequent compositions impedes rare compositions generation. We further show that existing compositional generation methods are ineffective at mitigating role collapse. (3) We propose ReBind, a simple framework that decomposes rare relations into effective intermediates, improving rare role generalization.

## 2  Related Works

**Text-to-Image Diffusion Models.** T2I diffusion models have made remarkable progress in generating high-quality images from text prompts [Betker et al., 2023, BlackForestLabs, 2024, Rombach et al.,

2022, Podell et al., 2023, Esser et al., 2024]. Stable Diffusion [Rombach et al., 2022] revolutionized the field by operating diffusion in the more efficient latent (rather than input) space and using CLIP [Radford et al., 2021] to encode text prompts. Subsequent models improved image quality and image-text alignment via architectural modifications [Esser et al., 2024], using multiple text encoders [Podell et al., 2023, Esser et al., 2024], or descriptive captions [Betker et al., 2023].

**Compositional Image Generation.** Text-to-image models often struggle with compositional generation, failing to bind objects, attributes, and relationships correctly [Zeng et al., 2023]. Recently, many approaches have been developed to mitigate this problem. Attention-based methods [Chen et al., 2024, Feng et al., 2023a, Chefer et al., 2023] modify the text-image cross-attentions to control the generation of objects, some use scene-graph to guide generation [Shen et al., 2024, Farshad et al., 2023], and others focus on improving spatial and relational correctness [Li et al., 2023, Yang et al., 2023, Lian et al., 2024, Yang et al., 2023]. Several works utilize LLM to improve prompt quality by providing context and/or conditions for compositional generation [Wu et al., 2024a,a, Yang et al., 2024, Feng et al., 2023b, Hu et al., 2024]. For instance, SLD [Wu et al., 2024a] leverages an LLM to identify errors (e.g. wrong object attribute) and suggest simple modifications (i.e., add, delete, or repositioning). RPG [Yang et al., 2024] is a re-captioning and planning method that first identifies objects in the prompt and generates their corresponding subregion. Although these approaches improve *spatial or attribute-based composition*, they often require complex external modules such as LLM, SAM [Kirillov et al., 2023], and object detector [Wu et al., 2024a, Feng et al., 2023b]. They do not explicitly resolve the inherent bias, and they often fail to control more challenging aspects such as orientation and facial expression, limiting their usability to generate complex active relations. Unlike these works, this paper specifically examines how *role priors* distort action-based generation and aims to improve action-based relation *without using any complex methods or additional prior information*: we do not manipulate the architecture or attention, nor additional (possibly costly) information such as bounding boxes, scene graphs, and depth.

**Rare Concept Generalization.** T2I models struggle to generalize to rare compositions that appear infrequently in training data. Recent works have primarily focused on static rare concept generalizations. [Samuel et al., 2024] addressed rare object categories (e.g., 'hand palms'), while [Cong et al., 2025, Park et al., 2025, Zhuang et al., 2024] focused on rare attribute-centric compositions within or across entities (e.g., "man wearing earrings" or "furry frog"). In contrast, we study directional action-based relations involving **rare subject–object interactions** – a more challenging setting beyond attribute or static generalization, as it requires accurate role binding and an understanding of directional dynamics in actions. Wu et al. [2024b] propose RRNet for relation rectification, studying similar action-based relations. However, it focuses on frequent interactions and fails to generalize to rare compositions where T2I models are prone to role collapse. Other work takes a more data-driven approach. [Okawa et al., 2023] study compositional generalization for simple attributes (color, size, and shape), concluding that generalization occurs when target compositions are structurally similar to training instances. [Chang et al., 2024] analyze how data distribution skews affect generation through "completeness" and "balance" metrics, but attribute failures to rarity alone, and do not specifically address the challenge of rare action relationships. Our work introduces a *novel observation*: T2I models can generate some rare compositions (e.g., "mouse chasing boy") while consistently failing on others (e.g., "mouse chasing cat") *when the inverse relationship is common in training data*.

# 3 Methodology

## 3.1 Background

**Diffusion Process.** Diffusion models [Rombach et al., 2022, Podell et al., 2023] transform a normal distribution into a data distribution via a series of denoising steps. Starting with a clean image $x_0$ and adding noise over time-steps $t \in [0, T]$ creates a noisy version $x_t = \sqrt{\alpha_t} x_0 + \sqrt{1 - \alpha_t} \epsilon$, where $\epsilon \sim \mathcal{N}(0, I)$ and $\alpha_t$ represents the noise schedule. For efficiency, this process typically operates in a latent space $z = \mathcal{E}(x)$ rather than pixel space.

**Training.** A neural network $\epsilon_\theta$ is trained to predict the noise added at each step, conditioned on the noisy input $z_t$ and a text prompt embedding $p$. The objective minimizes:

$$\mathcal{L}(\theta) = \mathbb{E}_{(x_0, p) \sim \mathcal{D}, \epsilon, t} \left[ \| \epsilon - \epsilon_\theta(z_t, p, t) \|_2^2 \right] \tag{1}$$

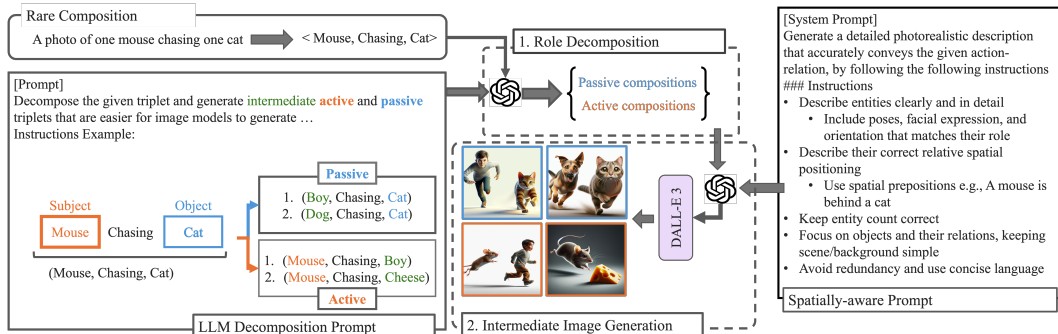

Figure 2: **Overview of ReBind.** Our method enhances rare composition generation by introducing structured e steps: (1) Role Decomposition via LLM-generated active/passive triplets to enforce correct role binding, and (2) Intermediate Image Generation using spatially-aware prompts. These images are later used for LoRA fine-tuning to mitigate role collapse.

**Inference.** During generation, we sample random noise $z_T \sim \mathcal{N}(0, I)$ and denoise it using the learned model. To enhance text-image alignment, classifier-free guidance [Ho and Salimans, 2021] combines conditional and unconditional predictions:

$$\tilde{\epsilon}_\theta(z_t, p, t) = (1 + w) \cdot \epsilon_\theta(z_t, p, t) - w \cdot \epsilon_\theta(z_t, \emptyset, t) \qquad (2)$$

where $w$ controls text conditioning strength.

### 3.2 Problem Definition

In compositional generation, prompts include sets of concepts $c = c_1, \ldots, c_n$, where each $c_i$ represents a sub-component like an object or attribute [Liu et al., 2022]. This work focuses on **directional action-based relations** formalized as triplets $c = (s, r, o)$, where subject $s$ performs action $r$ on object $o$ (e.g., "mouse chasing a cat"). A composition $c_R = (s_R, r, o_R)$ is rare when its reversed composition $c_F = (s_F = o_R, r, o_F = s_R)$ (i.e., "cat chasing mouse") appears much more frequently in the training data $\mathcal{D}$, i.e., $p_\mathcal{D}(c_F) \gg p_\mathcal{D}(c_R)$. We estimate $p_\mathcal{D}(\cdot)$ using LLMs' semantic plausibility, which correlate with data frequency [Kauf et al., 2024, Pedinotti et al., 2021].

We study this problem through RoleBench, a novel benchmark for evaluating role bias and rare compositional generation in action-based relations. Prior work studies low frequency of rare compositions Chang et al. [2024], while our unique focus is on how high frequency of reverse compositions specifically impedes rare generation. Our comprehensive analysis indicates that pre-trained T2I models exhibit strong role bias (Tab. 1) and often default to the *frequent* composition (e.g., "cat chasing mouse"). We term this behavior **role collapse**, where the model generates nearly identical images for both the rare and its frequent counterpart, formally $p_\theta(x \mid c_R) \approx p_\theta(x \mid c_F)$, and $\theta$ denotes the model parameters. Moreover, we observe that state-of-the-art compositional generation methods that use spatial priors (e.g., bounding box) or external modules (e.g., LLMs) are ineffective in mitigating role collapse (Tab. 2 and Fig. 5).

Our key insight is that overrepresentation of $c_F$ hinders $c_R$'s successful generation. We hypothesize that effective intermediate compositions that preserve $c_R$'s role assignments while avoiding role collapse can improve the model's generalization on $c_R$. We empirically test this via ReBind, a simple compositional framework. ReBind decomposes $c_R$ into multiple intermediates, denoted as $c_I$. We then fine-tune the model on these intermediates to enable the generation of $c_R$. We stress that this approach serves as a proof-of-concept rather than offering a general debiasing solution.

### 3.3 Reinforcing Role Binding through Directional Decomposition

We are seeking an answer to the question *Can rare compositions $c_R$ be decomposed into intermediate compositions that can be helpful to mitigate the directional bias in active relations?* To investigate, we propose **ReBind** framework, which leverages LLMs to strategically decompose the *rare composition* $c_R$ into *active* and *passive* intermediates to improve the compositional generalization of a T2I diffusion model through simple finetuning. Fig. 2 illustrates an overview of ReBind.

**Role Decomposition**. To help the model learn a rare composition $c_R = (s_R, r, o_R)$ (e.g., "mouse chasing cat"), we decompose it into two directional intermediates: an *active* composition $c_I^{\text{active}} = (s_R, r, o')$ and a *passive* composition $c_I^{\text{passive}} = (s', r, o_R)$, where $s', o' \notin \{s_R, o_R\}$. These preserve one role of $c_R$ while replacing the other with another object. We conjecture that these intermediates separately reinforce uncommon role bindings (e.g., mouse as chaser, cat as target), enabling the model to reconstruct the full rare relation more faithfully during generation. As shown in Fig. 2, we use in-context learning with a Chain-of-Thought (CoT) prompting strategy in an LLM to generate diverse, high-quality $s'$ and $o'$ (see Sec. C for full prompt and examples). To ensure the generated intermediates are effective, we impose two criteria: (1) **Low directional bias**: the intermediate must not suffer from the same role bias as $c_R$. In practice, we observe that selecting objects (i.e., agent and patient roles) from different categories (e.g., human, animals, or non-living objects) can be effective to satisfy this property. For instance, "mouse chasing *dog*" (both agent and patient are animals) is a poor choice because "dog chasing mouse" is more likely to be frequent. On the other hand, "mouse chasing *boy/cheese*" (agent is animal, patient is human/non-living object) is more appropriate as the reverse counterpart ("boy/cheese chasing mouse") is *less likely* to be frequent. (2) **Plausibility**: the substituted objects must support the intended action as such examples are likely to be rare in the data (e.g., "mouse chasing cheese" is valid, but "cloud holding cat" is not; refer to Sec. B for more details).

**Intermediates Generation**. We use DALL-E 3 to generate $N$ images for each $c_I^{\text{active}}$ and $c_I^{\text{passive}}$. We expand prompts with an LLM to improve generation quality (e.g., spatial cues, object configuration). We further keep only images that resemble high alignment with the original prompt (i.e., alignment more than a threshold). However, in practice, we observe that alignment scores do not accurately represent directional alignment (i.e., mouse chasing cat vs. cat chasing mouse) because existing MLLM-based metrisacs like VQAScore [Lin et al., 2024] provide global image-text alignment, failing to effectively represent directional alignment. Hence, we filter images based on the metric beta defined in Sec. 4. Specifically, we keep images with $\beta < 0$ capturing directional alignment.

**Fine-tuning Objective**. Let $\mathcal{D}_{\text{inter}}$ be the accepted intermediate image–prompt pairs. We fine-tune the T2I model using Low-Rank Adaptation (LoRA) [Hu et al., 2022] with the following objective:

$$\mathcal{L}_{\text{compos}} = \lambda \cdot \mathcal{L}_{\text{active}} + \mathcal{L}_{\text{passive}} \tag{3}$$

where $\mathcal{L}_{\text{active}}$ and $\mathcal{L}_{\text{passive}}$ are standard diffusion reconstruction losses (Eq. 1) applied to the active and passive intermediates, respectively. The coefficient $\lambda$ balances their influence. Since modeling active roles (e.g., "mouse chasing") is often more difficult, we apply a slow exponential ramp-up on $\lambda$ during training (see Sec. 5). Note that *spatially-aware prompt* (Fig. 2) is only used to generate intermediate images. During fine-tuning and evaluation, we use simple prompts.

## 4   Benchmark & Experimental Setup

**Benchmark.** RoleBench includes 10 common action-based relations with clear subject-object interactions: *chasing, riding, throwing, holding, following, feeding, pulling, lifting, carrying*, and *kissing*. For each action relation $r$, we define (1) *Frequent ($s_F, r, o_F$)* (e.g., "cat chasing mouse"), and (2) *Rare ($s_R, r, o_R$)* with reversed roles (e.g., "mouse chasing cat"), with $s_R = o_F$ and $o_R = s_F$. The goal is to evaluate how well T2I models can represent rare compared to frequent compositions. Since recent T2I models use closed-source training data, we employ LLM semantic plausibility as a proxy for frequency patterns, which has been shown to correlate with training data distributions [Kauf et al., 2024, 2023, Pedinotti et al., 2021]. We restrict RoleBench to animated objects to avoid impossible interactions (e.g., "chair kissing boy") and use a hybrid approach combining [Wu et al., 2024b] and GPT-4o generation with manual filtering. We validate frequent/rare compositions in RoleBench quantitatively: LLM log-probabilities show frequent compositions achieve higher plausibility in 80% of cases, and Google Search counts reveal an average rare-to-frequent ratio of 0.21, which is consistent with our analysis T2I models on RoleBench. More details are in Sec. B.

RoleBench includes two prompt types: (1) *basic prompts ("A photo of one {s} {r} one {o}")* and (2) *spatially-aware prompts* with detailed spatial descriptions (see Fig. 2). For each relation, we create 1 basic prompt and 20 spatially-aware prompts for both rare and frequent directions, totaling 42 prompts per relation (21 per direction) and 420 prompts overall. For each T2I model, we generate images for both directions: 20 images from the basic prompt and 20 from spatially-aware prompts (1 per prompt) per direction, yielding 40 images per direction per relation, or 800 images total per

model (40 images × 10 relations × 2 directions). 1 per each spatially-aware prompt), yielding 800 images per model (40 images x 10 relations x 2 role orders).

**Baselines.** Our goal is to assess directional role bias in action-based relations (i.e, a compositional generalization problem). We benchmark two types of baselines: (1) pre-trained T2I diffusion models: Stable Diffusion XL (SDXL) [Podell et al., 2023], DALL-E 3 [Betker et al., 2023], SD3 and SD3.5 [Esser et al., 2024], and AuraFlow2 [Fal, 2024]; (2) training-free compositional generation methods: LLM guided/feedback compositional models (SLD [Wu et al., 2024a] and RPG [Yang et al., 2024]); (3) training-based compositional method: Composition-Aware feedback optimization (IterComp) [Zhang et al., 2024]; (4) rare concept generation: R2F [Park et al., 2025]; and (5) relation-aware methods: RRNet [Wu et al., 2024b]. Most methods use SDXL as the base model, with RRNet using Stable Diffusion 2.1 (w=0.6) per original setting. We select SDXL as our backbone for fair comparison. To evaluate ReBind, we implement a Freq/Rare baseline (FT Freq/Rare) that directly fine-tunes SDXL on frequent and rare DALL-E 3 images. Results show that, in contrast to our method (i.e., ReBind), FT Freq/Rare is incapable of enhancing generalization on rare compositions and still defaults to the frequent compositions. We experiment with two active role hyper-parameter $\lambda$ settings: fixed ($\lambda = 1$) and gradually increasing ($\lambda = 2/5$), which exponentially ramps up from 0.5. In Sec. C, we demonstrate that ReVersion [Huang et al., 2024] fails to generalize to action-based relations. Preference-optimization methods [Huang et al., 2024, Ruiz et al., 2023, Gal et al., 2023] learn specific styles from a few examples, but suffer from existing compositional biases in models.

**Automated Evaluation.** We evaluate generated images using VQAScore [Lin et al., 2024] with 'clip-flant5-xxl' backbone, the current state-of-the-art compositional evaluation model. We include overall prompt alignment (standard VQAScore) and role-specific criteria: *spatial configuration*, *orientation*, *facial expression (gaze)*, and *pose*. These fine-grained scores effectively evaluate **role binding** beyond overall alignment. Inspired by [Cho et al., 2024], we generate binary questions for each category using GPT-4o (e.g., "Is the mouse positioned in front of the cat?"), where the expected answer is "yes" if the image correctly represents the intended composition. Each category score is the average VQAScore across its questions. Prompts in to Sec. F.

**Bias $\beta$.** Image-Text Alignment (ITA) metrics like VQAscore measure alignment between image and text. However, these metrics fail to capture fine-grained details, such as direction, orientation, and spatial configuration. In addition to our specific role-specific criteria (see above), to better capture the direction of the action (i.e. mouse→chasing→cat vs. cat→chasing→mouse), we define **role direction bias $\beta$**. $\beta$ measures the degree to which the generated image matches the reversed composition over the intended composition. Formally, $\beta = -\Delta$ where $\Delta = \text{VQAScore}(I, p_c) - \text{VQAScore}(I, p_{c^{rev}})$, with $I$ and $p_c$ denoting image and prompt for composition $c$, and $p_{c^{rev}}$ denoting the reversed prompt. We refer to $\text{VQAScore}(I, p_c)$ and $\text{VQAScore}(I, p_{c^{rev}})$ as *matching* and *unmatching* scores, respectively. A positive $\beta$ indicates bias towards the reversed composition, with lower values closer to 0 indicating less bias, and $\beta \leq 0$ indicates the model is not biased and the generated image is well-aligned with the intended composition. T2I models exhibit positive $\beta$ for rare compositions (Tab. 1).

**Human Evaluation.** To complement automated metrics, we conduct human evaluations where six annotators per task compare top-3 VQAScore images from our method against six baselines. Annotators rate prompt alignment and relation quality in pairwise comparisons, and rank images from best to worst. Baselines generated potentially inappropriate entangled objects for "kissing." We therefore excluded this relation from human evaluation and qualitative analysis. Additionally, we collect human annotations for DALL-E 3 images, with annotators rating alignment to both frequent and rare prompts on a 1-5 scale. In both studies, we use Amazon Mechanical Turk with quality checks to ensure reliable annotations.

**Implementation Details.** We fine-tune models using LoRA (rank 32) for 1,000 steps on an L40S GPU with a 1e-4 learning rate and bf16 precision. For each rare triplet, we generate 2–3 active/passive triplets using GPT-4o. We fix the seed to 24 for training and ablations, but we do not use a seed for generating RoleBench to ensure output variability. We set $N = 20$. More details are in Sec. E.

## 5 Results

This paper aims to study directional role bias in T2I models through action-based relations. First, we demonstrate that state-of-the-art models fail to generate rare compositions like "mouse chasing

Table 1: **Pre-trained T2I models exhibit strong role bias**. T2I models show strong directional bias, reflected in consistently positive $\beta =$ Unmatch $-$ Match. *Match* denotes alignment between image and the original prompt; *Unmatch* evaluates alignment with the reversed prompt. All scores are in %. ↓ Smaller $\beta$ is better. Lowest $\beta$ Freq/Rare is bolded. Type refers to relation type.

| Type | T2I Model | Size (B) | Spatial | | | Orientation | | | Pose | | | Facial Expression | | | VQAScore | | |
|---|---|---|---|---|---|---|---|---|---|---|---|---|---|---|---|---|---|
| | | | M | U | $\beta\downarrow$ | M | U | $\beta\downarrow$ | M | U | $\beta\downarrow$ | M | U | $\beta\downarrow$ | M | U | $\beta\downarrow$ |
| **Freq** | SDXL | 3.5 | 82.7 | 67.8 | -14.9 | 76.1 | 68.6 | -7.5 | 74.2 | 69.1 | -5.1 | 81.8 | 70.3 | -11.5 | 84.0 | 57.7 | -26.3 |
| | SD3 | 2 | 80.2 | 65.1 | -15.1 | 74.1 | 68.9 | -5.2 | 71.5 | 64.2 | -7.3 | 78.4 | 68.2 | -10.2 | 82.6 | 53.3 | -29.3 |
| | SD3.5 | 2.5 | 83.3 | 66.1 | -17.2 | 76.1 | 68.9 | -7.2 | 73.0 | 64.7 | -8.3 | 81.2 | 69.6 | -11.6 | 84.9 | 51.3 | **-33.6** |
| | AuraFlow2 | 6.8 | 87.2 | 69.1 | **-18.1** | 78.5 | 71.3 | -7.2 | 76.9 | 69.1 | -7.8 | 83.2 | 70.4 | **-12.8** | 84.4 | 60.8 | -23.6 |
| | DALL-E 3 | 12 | 86.9 | 70.5 | -16.4 | 78.3 | 70.2 | **-8.1** | 77.5 | 67.2 | **-10.3** | 80.6 | 67.8 | **-12.8** | 88.3 | 60.6 | -27.7 |
| **Rare** | SDXL | 3.5 | 68.2 | 80.3 | 12.1 | 68.6 | 74.3 | 5.7 | 70.3 | 73.9 | 3.6 | 71.7 | 80.0 | 8.3 | 56.6 | 79.7 | 23.1 |
| | SD3 | 2 | 65.1 | 75.4 | 10.3 | 66.4 | 70.3 | **3.9** | 64.8 | 66.4 | **1.6** | 68.5 | 75.3 | 6.8 | 56.4 | 73.6 | 17.2 |
| | SD3.5 | 2.5 | 66.9 | 77.8 | 10.9 | 68.3 | 72.6 | 4.3 | 65.9 | 69.0 | 3.1 | 72.7 | 79.1 | 6.4 | 59.5 | 80.4 | 20.9 |
| | AuraFlow2 | 6.8 | 72.9 | 83.7 | 10.8 | 71.8 | 77.6 | 5.8 | 75.1 | 77.9 | 2.8 | 76.6 | 83.6 | 7.0 | 74.6 | 84.7 | **10.1** |
| | DALL-E 3 | 12 | 74.9 | 84.1 | **9.2** | 71.3 | 75.6 | 4.3 | 73.0 | 76.5 | 3.5 | 73.6 | 78.1 | **4.5** | 68.9 | 84.9 | 16.0 |

cat," often **defaulting to their frequent counterparts** (i.e., they exhibit role collapse) (Table 1). Second, we show that existing compositional generation methods are ineffective at mitigating role collapse (Table 2). Third, we test our hypothesis through ReBind by evaluating whether intermediate compositions (i.e., active/passive) are effective to mitigate role collapse and enable T2I models to generate rare compositions (Fig. 3 and Table 2). Fig. 5 compares ReBind qualitatively and Sec. 5.2 analyzes ReBind through effective ablation studies. Our goal is not to introduce a new T2I model, but to understand why current models fail on rare directional compositions and whether a simple decomposition strategy improves generalization.

## 5.1 Main Results

**Pre-trained T2I models exhibit strong directional role bias**. Table 1 shows that all pre-trained T2I models exhibit strong role directional bias when generating rare action-based compositions. We measure this bias using $\beta =$ Unmatch $-$ Match: a higher $\beta$ indicates the model aligns more with the *reversed* prompt than the intended prompt. Across all models and evaluation dimensions, rare compositions consistently yield positive $\beta$, confirming a **strong preference for the frequent counterpart** (i.e., role collapse). For instance, when prompted with a "mouse chasing cat," models often depict "cat chasing mouse".

This bias is especially strong in spatial positioning, but extends across all categories. Frequent compositions, in contrast, yield strongly negative $\beta$, indicating correct role alignment. The gap between rare and frequent is often over 40 points in VQAScore $\beta$, highlighting the severity of this collapse. Notably, larger models (e.g., DALL-E 3, AuraFlow2) outperform smaller ones on frequent prompts, but show similar or even worse bias ($\beta$) on rare ones. These results confirm that **even advanced T2I models default to learned priors** rather than following prompt semantics in rare cases, not due to model capacity, but due to **training-set dominance of frequent patterns**. Interestingly, the gap in Match scores between large and small models is much greater on rare prompts than on frequent ones, suggesting that **smaller models make broader generation errors** (e.g., object count, realism), while larger models still generate coherent scenes, yet with incorrect (reversed) roles.

**Compositional generation methods fail to mitigate role bias.** Table 2 shows that despite specialized architectures and training on compositional datasets, recent methods struggle with directional role bias. All models retain high $\beta$ values, indicating a consistent collapse toward frequent role patterns. IterComp and RPG offer only modest gains over SDXL compared to ReBind, with RPG showing **high spatial bias despite region-controlled generation**, highlighting that distributional priors, not architecture alone, drive the failure. Further, we observe that R2F and RRNet, methods designed for rare concept generation and action relation, respectively, are not effective, suggesting that current methods do not mitigate the existing bias in T2I models. SLD performs comparably to ReBind on individual criteria but shows significantly higher overall bias ($+8.7$ $\beta$). Note that SLD optimizes for VQAScore via image editing based on GPT-4o feedback, which can lead to superficial improvements (i.e., score hacking) rather than faithful generation.

Table 2: **Compositional generation methods are ineffective in reducing role bias.** All models exhibit a persistent role bias (positive $\beta$). ReBind achieves comparable or better bias reduction across all categories. RRNeT is trained using DALL-E 3 outputs as ground truth. M and U denote matching and unmatching scores, respectively. All scores in %. $\lambda = 5$. Lower $\beta$ indicates less bias. The best $\beta$ scores and those within 0.5 of the best are shown in bold. Green/Red indicate improvement/drop relative to SDXL.

| Model | Spatial | | | Orientation | | | Pose | | | Facial Expression | | | VQAScore | | |
|---|---|---|---|---|---|---|---|---|---|---|---|---|---|---|---|
| | M | U | $\beta\downarrow$ | M | U | $\beta\downarrow$ | M | U | $\beta\downarrow$ | M | U | $\beta\downarrow$ | M | U | $\beta\downarrow$ |
| DALL-E 3 | 74.9 | 84.1 | 9.2 - | 71.3 | 75.6 | 4.3 - | 73.0 | 76.5 | 3.5 - | 73.6 | 78.1 | 4.5 - | 68.9 | 84.9 | 16.0 - |
| SDXL | 68.2 | 80.3 | 12.1 - | 68.6 | 74.3 | 5.7 - | 70.3 | 73.9 | 3.6 - | 71.7 | 80.0 | 8.3 - | 56.6 | 79.7 | 23.1 - |
| IterComp | 66.8 | 80.3 | 13.5 (-1.4) | 68.4 | 75.3 | 6.9 (-1.2) | 68.5 | 72.5 | 4.0 (-0.4) | 70.0 | 80.3 | 10.3 (-2.0) | 59.2 | 81.4 | 22.2 (0.9) |
| RRNet | 68.2 | 76.8 | 8.6 (3.5) | 67.5 | 71.2 | 3.7 (2.0) | 69.1 | 72.2 | **3.1** (0.5) | 71.0 | 74.7 | 3.7 (4.6) | 61.2 | 76.3 | 15.1 (8.0) |
| R2F | 68.0 | 79.4 | 10.7 (1.4) | 68.0 | 73.4 | 5.4 (0.3) | 71.0 | 74.4 | 3.4 (0.2) | 71.9 | 79.3 | 7.4 (0.9) | 43.4 | 59.1 | 15.7 (7.4) |
| RPG | 65.4 | 74.3 | 8.9 (3.2) | 66.6 | 70.6 | 4.0 (1.7) | 63.7 | 66.7 | **3.0** (0.6) | 67.3 | 75.8 | 8.5 (-0.2) | 56.6 | 73.3 | 16.7 (6.4) |
| SLD | 68.4 | 75.1 | **6.7** (5.0) | 67.2 | 70.1 | **2.9** (2.8) | 67.2 | 70.0 | **2.8** (0.8) | 69.8 | 74.9 | 5.0 (3.3) | 56.8 | 73.4 | 16.6 (6.5) |
| **ReBind** | 68.5 | 75.5 | **7.0** (5.1) | 66.0 | 69.6 | 3.6 (2.1) | 73.3 | 76.5 | **3.2** (0.4) | 67.7 | 70.2 | **2.5** (5.8) | 57.9 | 65.8 | **7.9** (15.2) |

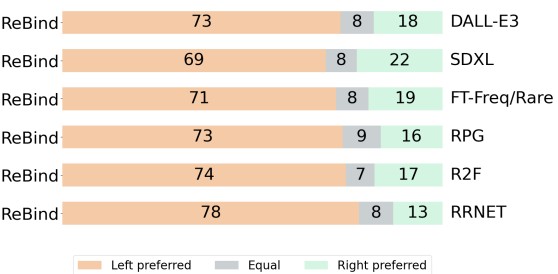

Figure 3: **ReBind is more effective than compositional generation methods**. Humans significantly prefer ReBind in a head-to-head comparison compared to baselines. Numbers in %.

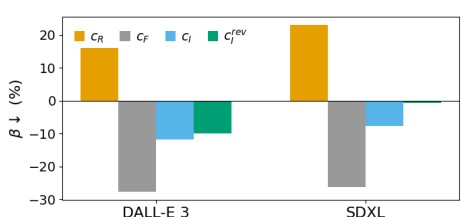

Figure 4: **Intermediates do not suffer from role bias**. Unlike rare compositions, intermediate compositions exhibit negative $\beta$.

**Passive/active compositions facilitate generalization to rare compositions through better role-binding.** Tab. 2 shows that ReBind significantly reduces SDXL bias across all criteria (i.e., both global score and target criteria)–e.g., 5.1 on spatial, 5.8 on facial expression, and 15.2 on global alignment. This supports that active/passive compositions help the model internalize role bindings and recover relation directionality to successfully generate rare compositions (examples in Fig. 5).

**ReBind is more effective than compositional generation methods**. As shown in Fig. 3, ReBind is consistently preferred by humans by a large margin. It outperforms DALL-E 3 and SDXL in 73% and 69% cases, respectively, and is strongly favored over RPG (73%), R2F (74%), and RRNet (78%). These results further confirm our findings. Table 2 further assesses ReBind against recent compositional generation, rare concept generation (R2F), and RRNeT (specifically designed to learn similar relations). ReBind achieves consistently lower $\beta$. For instance, it reduces bias by 8.8, 7.8, and 7.2 points more than RPG, R2F, and RRNet on VQAScore, respectively. However, unlike these specialized methods, it does not require region-controlled, heavy LLM usage, pre-layout conditioning, or external graphs as in RRNeT. In fact, ReBind reduces the spatial bias (spatial in Tab. 2) compared to most methods, while largely improving on non-spatial categories (e.g., 6,7.8, and 2.5 points compared to RPG, IterComp, and SLD, respectively).

## 5.2 Ablation

**Active/passive intermediates do not manifest role bias**. As shown in Fig. 4, intermediate compositions achieve negative $\beta$, indicating that they do not suffer from the role bias seen in rare compositions. For instance, DALL-E 3 and SDXL obtain $\beta = -11.8$ and $-7.7$ on intermediates (e.g., "mouse chasing boy"), respectively. This confirms correct action direction without defaulting to the reversed prompt ("boy chasing mouse"). Notably, Table 9 and Fig. 4 show that both intermediate and their

DALL-E 3        SDXL        R2F        RRNeT        RPG        ReBind

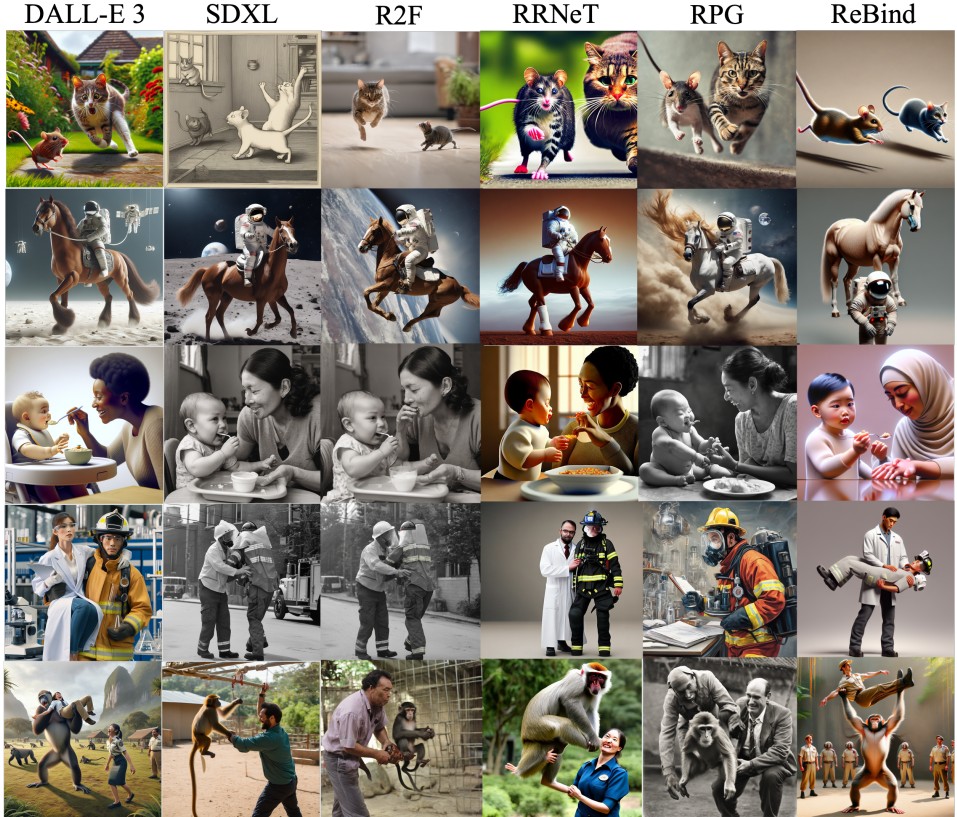

Figure 5: **Qualitative comparison of ReBind.** Baselines often collapse to frequent compositions, while ReBind better captures rare compositions. Top to bottom: mouse chasing cat, horse riding astronaut, boy feeding woman, scientist carrying fireman, monkey lifting zoo trainer.

Table 3: $\lambda$ **Ablation**. Fine-tuning with intermediate compositions reduces bias. Higher active weight $\lambda$ improves subject-role binding. M/U is matching/unmatching. All numbers are in %. ↓ smaller scores are better.

| Model | Intermediates | $\lambda$ | M | U | $\beta \downarrow$ |
|---|---|---|---|---|---|
| SDXL | – | – | 56.5 | 79.7 | 23.0 |
| FT Freq/Rare | – | – | 70.1 | 85.1 | 14.7 |
| ReBind | Active | 5 | 56.4 | 66.0 | 9.6 |
|  | Passive | – | 55.0 | 65.1 | 10.1 |
| ReBind | Active+Passive | 1 | 63.4 | 75.5 | 12.0 |
|  |  | 2 | 59.4 | 68.1 | 8.7 |
|  |  | 5 | 57.9 | 65.8 | **7.9** |

SDXL    FT Freq/Rare    ReBind ($\lambda$=1)    ReBind ($\lambda$=2)*    ReBind($\lambda$=5)*

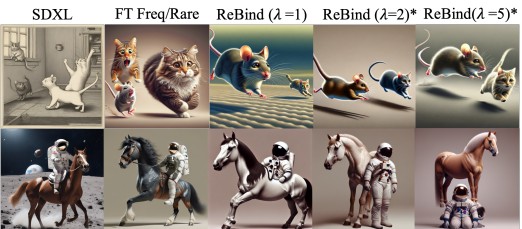

Figure 6: **Qualitative ablation.** Unlike ReBind, FT Freq/Rare is ineffective in mitigating role bias and still collapses to the frequent compositions. Higher $\lambda$ improves role-specific generation. * indicates a gradually increasing weight.

reversed counterpart (i.e, $c_I^{rev}$) exhibit small or negative $\boldsymbol{beta}$ (e.g., DALL-E 3: -11.8 vs. -9.9). This contrasts with rare/frequent pairs, where $\beta$ remains large and asymmetric.

$\lambda$ **ablation.** Table 3 demonstrates that active compositions are slightly more effective compared to passive compositions, with their combination achieving the best score (7.9) at $\lambda = 5$. Increasing the weight of active intermediates reduces role bias, improving the overall VQAScore by 4.1 points when $\lambda$ increases from 1 to 5. Active intermediates help overcome strong prior associations, enabling rare subjects to bind correctly to active roles. Figure 6 shows that higher weights allow subjects (e.g., mouse) to demonstrate proper chasing behavior with clear directional intent and positioning. Similarly, with a higher $\lambda$, the astronaut gradually assumes the correct position and role. These

Table 4: **ReBind maintains negative $\beta$ on frequent compositions**. $\beta$ scores in %; FE: Facial Expression, VS: VQAScore.

| T2I | Spatial | Orient. | Pose | FE | VS |
|---|---|---|---|---|---|
| SDXL | -14.9 | -7.5 | -5.1 | -11.5 | -26.3 |
| DALL-E 3 | **-16.4** | **-8.1** | **-10.3** | **-12.8** | **-27.7** |
| ReBind | -12.3 | -5.0 | -5.9 | -9.5 | -16.9 |

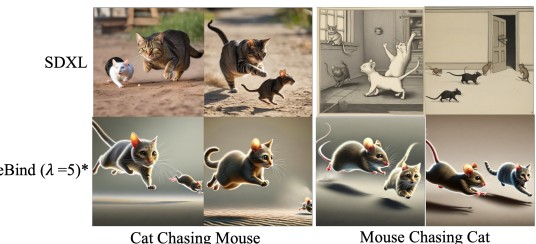

Figure 7: Frequent and rare visualization on chasing relation.

results confirm that active/passive intermediates effectively reduce bias and enable rare composition generation.

**Fine-tuning on Freq/Rare directly is ineffective.** To further evaluate the effectiveness of intermediate compositions and assess bias in T2I models, we directly train on Rare/Freq images generated by DALL-E 3 (i.e. "mouse chasing cat"/"cat chasing mouse"). We observe that ReBind is significantly preferred by humans (Fig. 3). Similarly, Fig. 6 shows FT Freq/Rare defaults to frequent composition due which is expected as both freq/rare prompts generate *frequent* composition.

**Generalization on Frequent Compositions.** ReBind shows improvement on rare compositions (e.g., "mouse chasing cat") using intermediate compositions This raises the question: *Does optimizing for rare directions harm the model's ability to generate frequent ones?* While performance drop is expected as ReBind fine-tunes on only rare directions, Tab. 4 shows that ReBind maintains a high negative $\beta$ on frequent composition. In Tab. 6 we observe that ReBind performs comparably to SDXL on the Matching score, with only small drops in matching scores: 0.5% (spatial), 1.7% (orientation), 2.4% (facial expression), and a 6.9% gain in pose. Also, Fig. 7 illustrates that the model can faithfully generate both freq/rare directions, while introducing visual artifacts such as object entanglement that cause image-text alignment to drop, specifically global VQAScore. More discussion in Table 6 and Fig. 10 in Sec. C. A more detailed discussion on limitations can be found in Sec.D.

**Limitations of Prompt Engineering for Rare Compositions.** Although strong T2I models like DALL-E 3 slightly benefit from spatially-aware prompting, it worsens results for SDXL and IterComp, likely due to their weaker language understanding capabilities (Fig. 11 in Sec. C).

# 6 Conclusion

This paper identifies a role-collapse, a failure mode where T2I models default to frequent compositions instead of faithfully generating rare compositions in action-based relations. Our key insight is that this failure stems from imbalanced distributional biases rather than fundamental limitations in model architecture. To test this hypothesis, we propose ReBind, which decomposes rare compositions into active and passive intermediate pairs that preserve role assignments while avoiding the dominant frequent composition. Through comprehensive experiments on RoleBench, we show the effectiveness of active/passive intermediates on mitigating role-collapse via a simple fine-tuning approach, corroborating our hypothesis. This work opens promising directions for improving compositional generalization in T2I systems, leading to more creative content generation.

# 7 Acknowledgment

This work is supported by NSF Grant No. 2329992. It is also supported by the University of Pittsburgh Center for Research Computing, RRID:SCR 022735, through the resources provided. Specifically, this work used the H2P cluster, which is supported by NSF award number OAC-2117681. We gratefully acknowledge the support of those who contributed to the human evaluation.

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

# A    Appendix Overview and Outline

This appendix provides extended results, implementation details, and additional analysis to support the main paper and investigate the following questions:

- Can T2I diffusion models generate rare directional relations (e.g., "mouse chasing cat") faithfully?
- Are failures due to model limitations or scarcity in training data?
- Can compositional methods address these failures?
- Does decomposing rare relations into directional intermediates improve generalization?

This Appendix is organized as follows:

- **Benchmark details** (Sec. B): An overview of RoleBench, including relation categories, frequent/rare compositions, and the intermediate prompts generated by ReBind (Table 5).
- **Extended results and analysis** (Sec. C):
    - Fig. 8 shows role bias through attention maps
    - Fig. 6: Additional qualitative examples.
    - Table 6 and Fig. 10: Results on frequent compositions and comparisons to rare counterparts.
    - Table 7: Evaluation of generalization in multi-relation training.
    - Table 8 Evaluation on coco prompts.
    - Table 9: Performance on intermediate prompts and their reversed variants.
    - Table 10: Ablation on $N$.
    - Fig. 11: Effects of spatially-aware prompt engineering.
    - Fig. 12: Limitations of ReVersion Huang et al. [2024] on our benchmark.
- **Limitations and future directions** (Sec. D): Analysis of remaining challenges, including multi-object relations and generalization gaps (Fig. 13).
- **Experimental setup** (Sec. E): Implementation details including model configuration, hyperparameters, and training strategy.
- **Prompt templates and examples** (Sec. F): Full set of prompts used for generation and examples used in-context.
- **Human evaluation illustrations**: Examples from the human preference study is included in the supplementary.

# B    RoleBench Benchmark

## B.1    Data Construction

We introduce **RoleBench**, a benchmark for evaluating whether T2I diffusion models can faithfully generate rare, directional, action-based compositions. As detailed in Tab.5, RoleBench covers 10 common action-based relations. In order to effectively analyze directional role bias, we focus on actions that have a clear distinction between each direction (e.g., "A chasing B"vs "B chasing A" is clear, while "A fighting B" vs "B fighting A" is not). For each action relation $r$, we define:

- A *frequent* composition $x_F = (s_F, r, o_F)$, where the subject $s_F$ and object $o_F$ follow typical role priors.
- A corresponding *rare* composition $x_R = (s_R, r, o_R)$, where $s_R = o_F$ and $o_R = s_F$, reversing the role assignments.

Since the training data of recent T2I models is not publicly available, we adopt semantic plausibility as a proxy for frequency patterns. Semantic plausibility (knowing that "cat chasing mouse" is more plausible than "mouse chasing cat") effectively captures world knowledge and correlates with training data frequency Kauf et al. [2024], Pedinotti et al. [2021], Kauf et al. [2023]. Since LLMs and T2I

Table 5: **RoleBench.** RoleBench includes 10 action-based relations. We construct frequent and rare compositions and decompose the rare composition into 2-3 active/passive intermediate pairs. Note that all compositions (subject, relation, object) except for *throwing* can be read as *A photo of one {subject} {relation} one{object}* (e.g., A photo of one cat chasing one mouse). *Throwing* is a special case including 3 objects. i.e., (boy, throwing, puppy) reads *A photo of one boy throwing **a ball** to one puppy*

| Relation | Frequent Case | Rare Case | Active Intermediate | Passive Intermediate |
|---|---|---|---|---|
| Chasing | (cat, chasing, mouse) | (mouse, chasing, cat) | (mouse, chasing, boy), (mouse, chasing, cheese) | (dog, chasing, cat), (boy, chasing, cat) |
| Riding | (astronaut, riding, horse) | (horse, riding, astronaut) | (horse, riding, bear), (horse, riding, dog) | (bear, riding, astronaut), (dog, riding, astronaut) |
| Throwing | (boy, throwing, puppy) | (puppy, throwing, boy) | (puppy, throwing, stick), (puppy, throwing, cat) | (robot, throwing, boy), (girl, throwing, boy) |
| Holding | (grandpa, holding, doll) | (doll, holding, grandpa) | (doll, holding, baby), (doll, holding, teddy bear) | (man, holding, grandpa), (monkey, holding, grandpa) |
| Following | (lion, following, cow) | (cow, following, lion) | (cow, following, farmer), (cow, following, truck) | (person, following, lion), (dog, following, lion) |
| Feeding | (woman, feeding, baby) | (baby, feeding, woman) | (baby, feeding, teddy bear), (baby, feeding, cat) | (man, feeding, woman), (robot, feeding, woman) |
| Kissing | (mother, kissing, baby) | (baby, kissing, mother) | (baby, kissing, teddy bear), (baby, kissing, doll) | (daughter, kissing, mother), (father, kissing, mother) |
| Pulling | (man, pulling, dog) | (dog, pulling, man) | (dog, pulling, sled), (dog, pulling, cart) | (boy, pulling, dog), (woman, pulling, dog) |
| Lifting | (zoo trainer, lifting, monkey) | (monkey, lifting, zoo trainer) | (monkey, lifting, robot), (monkey, lifting, box) | (robot, lifting, zoo trainer), (gorilla, lifting, zoo trainer) |
| Carrying | (fireman, carrying, scientist) | (scientist, carrying, fireman) | (scientist, carrying, robot), (scientist, carrying, child), (scientist, carrying, backpack) | (gorilla, carrying, fireman), (robot, carrying, fireman), (elephant, carrying, fireman) |

models share similar web-based training data, LLM plausibility estimates provide proxies for T2I model biases, as well as generating and validating the selected frequent and rare compositions.

We restrict RoleBench to animated objects because inanimate objects create impossible relations Kauf et al. [2023] (e.g., "chair kissing person"), and employ a semi-automatic approach to acquire frequent and rare compositions. We created candidate pairs by combining related works data Wu et al. [2024b] and prompting GPT-4o to generate additional pairs where one direction is plausible (frequent) and the reverse is implausible (rare). We then manually filter to ensure proper structure and select one pair per relation, ensuring object diversity (e.g., not using mouse/cat for all relations). We stress that these design choices are adopted according to the main goal of this paper, i.e., a controlled study on directional role bias and testing compositional generalization of T2I models, rather than providing another large-scale benchmark.

RoleBench comprises two prompt types for each composition: (1) *basic prompt* following "A photo of one {s} {r} one {o}", and (2) *spatially-aware prompt* with detailed spatial descriptions. For each composition, we include 1 basic and 20 spatially-aware prompts, totaling 42 prompts per relation and 420 overall. We generate 40 images per relation per model (20 basic, 20 spatially-aware), yielding 800 images per model across all relations and role orders.

## B.2   Data Validation

We further conduct quantitative analysis to ensure the correctness of the chosen frequent and rare compositions. However, similar to generation, the caveat is that recent T2I models use closed-source training data, making direct statistical analysis infeasible. We therefore employ established proxy methods that have been validated in recent literature. (1) **LLM log-probability estimation**: Following Kauf et al. [2024], we use Mistral-7B's log-probabilities as a proxy for training data frequency through semantic plausibility estimation. Specifically, we estimate $p_{\mathcal{D}}(c_F)$ and $p_{\mathcal{D}}(c_R)$ for frequent and rare compositions, respectively, validating our selections when $p_{\mathcal{D}}(c_F) - p_{\mathcal{D}}(c_R) > 0$. Our analysis shows frequent compositions achieve higher plausibility in 80% of generated compositions. (2) **Google Search frequency counts**: We validate composition frequency using web search counts, which reveal an average rare-to-frequent ratio ( count($c_F$) / count($c_R$)) to be 0.21. These results align with our T2I benchmark evaluation, where models consistently performed better on frequent compositions compared to rare compositions, confirming the validity of our composition categorization.

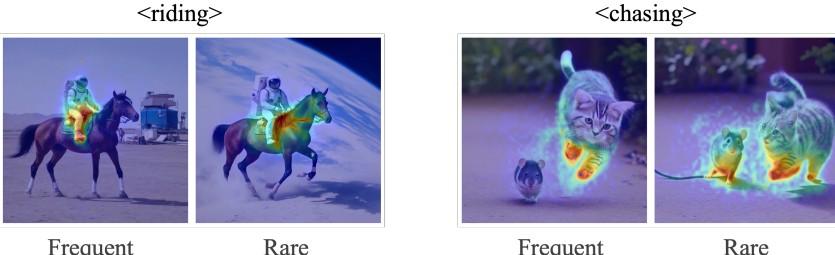

Figure 8: **Attention maps on frequent and rare compositions.** Visualization of cross-attention for the action token (`<riding>` and `<chasing>`). For frequent compositions, attention focuses on the correct agent (e.g., astronaut or cat). For rare compositions, attention becomes diffused or incorrectly spans both subject and object, indicating role confusion. All images are generated with SD3.5-medium.

### B.3 Human Evaluation Details

We conduct two human evaluation studies to assess the generation quality better and validate alignment metrics.

(1) In the first study, we ask 3 annotators to evaluate 5 examples per relation generated by DALL-E 3 using spatially-aware prompts. Each annotator is shown a frequent and a rare composition (randomized order) and asked to rate each image on a 1–5 scale for its alignment with the "matching" and "unmatching" prompts. Annotators also indicate which image better reflects the intended frequent or rare prompt, and specify reasons for assigning a score less than 5.

To compare with automated metrics, we compute Pearson's correlation between human ratings and VQAScore-derived metrics. We observe $r_{\text{matching}} = 0.4675$, $r_{\text{unmatching}} = 0.369$, and $r_\beta = 0.491$, confirming that $\beta$ better reflects directional role agreement. In contrast, the unmatching score appears less informative. Therefore, we prioritize matching and $\beta$ scores in our main analysis.

(2) In the second study, we compare ReBind against six baselines: DALL-E 3, SDXL, RPG, RRNeT, R2F, and FT Freq/Rare. For each model, we select the top-3 generations per prompt based on VQAScore. Annotators then perform pairwise comparisons across all model combinations, choosing the image that better matches the rare composition. Each pairwise comparison is rated by 6 annotators. An illustration of both studies is included at the end of the Appendix.

## C   Results

**Role bias through attention maps**. Figure 8 visualizes cross-attention maps for action tokens in frequent and rare compositions. We observe that for frequent cases (e.g., "cat chasing mouse"), attention concentrates on the correct agent, accurately capturing the subject performing the action. In contrast, for rare compositions ("mouse chasing cat"), attention becomes dispersed or misaligned, often covering both entities. This suggests that models conflate agent and patient roles under distributional asymmetry, leading to the observed role-collapse.

**Qualitative ablation.** Fig. 6 shows generations for two rare compositions: "mouse chasing cat" and "horse riding astronaut." SDXL and FT Freq/Rare often collapse to the frequent counterpart (e.g., "cat chasing mouse") or produce ambiguous scenes with duplicated objects. **This confirms the strong bias of pretrained models and shows that direct fine-tuning on rare prompts often reinforces dataset priors**. In contrast, ReBind consistently improves role binding. For "mouse chasing cat," increasing $\lambda$ leads to clearer spatial layout, better object orientation, and more expressive actions (e.g., leaning). Although "horse riding astronaut" remains difficult, ReBind avoids collapsing to the frequent case and encourages more plausible poses (e.g., the astronaut sitting) as $\lambda$ increases. These results support our hypothesis that directional intermediates guide models toward better rare composition generation.

**Intermediate fine-tuning reduces bias without harming frequent generation.** Table 6 and Fig. 10 show that all models achieve negative $\beta$ on frequent prompts (e.g., "cat chasing mouse"), indicating

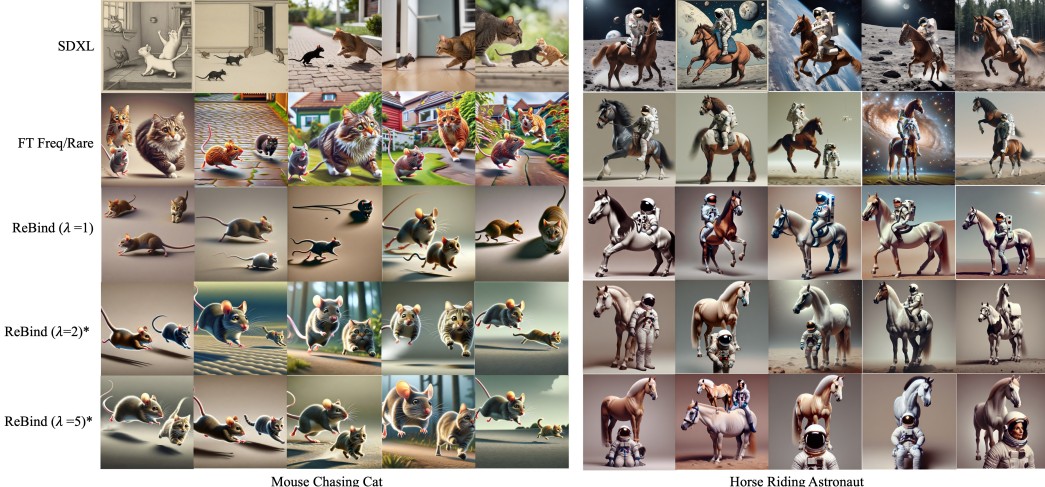

Figure 9: **Qualitative ablation on** $\lambda$. ReBind mitigates directional role biases in T2I models, whereas FT Freq/Rare overfits to dataset biases. Higher active weight $\lambda$ improves subject-role binding. $*$ weight is gradually increased.

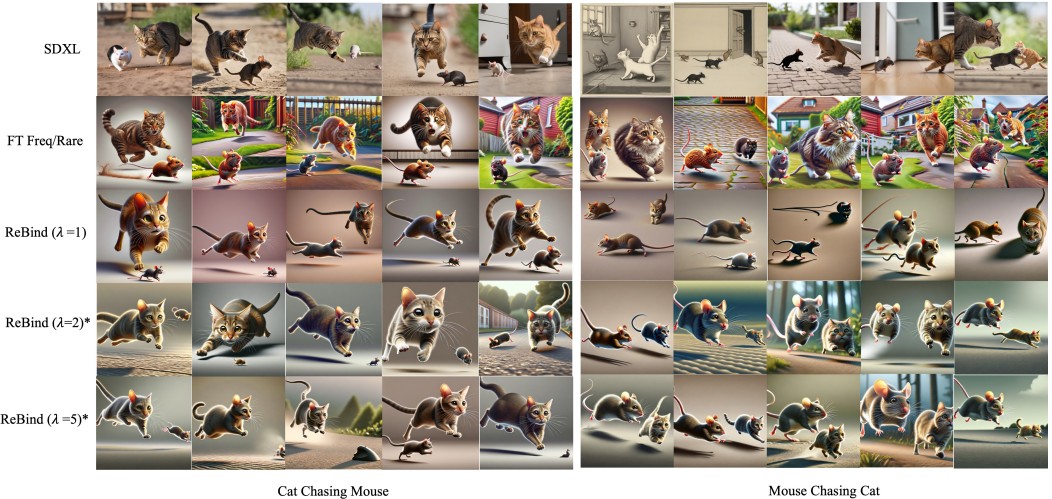

Figure 10: **Qualitative ablation on Frequent vs. Rare**.

strong alignment with the intended direction. ReBind ($\lambda = 5$) performs comparably to SDXL, with only small drops in matching scores: 0.5% (spatial), 1.7% (orientation), 2.4% (facial expression), and a 6.9% gain in pose. This suggests that fine-tuning on intermediate prompts preserves performance on frequent cases while improving rare composition generation (see Table 2 in the main paper).

Fig. 10 also highlights that FT Freq/Rare overfits, producing frequent compositions even on rare prompts. In contrast, ReBind maintains directional intent across both rare and frequent prompts. While its unmatching scores may be slightly higher than SDXL's, this reflects improved object depiction, not confusion in role direction, since rare and frequent prompts often differ only by subject-object roles. Overall, ReBind reduces role-collapse and enables better generalization without sacrificing accuracy on frequent cases.

**Generalization under Multi-Relation Training.** Unlike RRNeT [Wu et al., 2024b], which trains a separate model per composition, we evaluate whether a single model trained across multiple relations can generalize to unseen subject–relation–object compositions. Specifically, we train ReBind using intermediate compositions from six relations: *chasing*, *riding*, *lifting*, *pulling*, *holding*, and *throwing*. Evaluation covers four categories: (1) **In-Domain** compositions (relation and objects) entirely seen

Table 6: **ReBind doesn't significantly hurt performance on frequent compositions.** Each column shows Matching, Unmatching, and $\beta = \text{Unmatch}(U) - \text{Match}(M)$ for five axes: Spatial, Orientation, Pose, Facial Expression, and VQAScore. Negative $\beta$ indicates better separation between aligned and misaligned generations. All values in %.

| T2I Model | Spatial | | | Orientation | | | Pose | | | Facial Expression | | | VQA Score | | |
|---|---|---|---|---|---|---|---|---|---|---|---|---|---|---|---|
| | M | U | $\beta\downarrow$ | M | U | $\beta\downarrow$ | M | U | $\beta\downarrow$ | M | U | $\beta$ | M | U | $\beta\downarrow$ |
| SDXL | 82.7 | 67.8 | -14.9 | 76.1 | 68.6 | -7.5 | 74.2 | 69.1 | -5.1 | 81.8 | 70.3 | -11.5 | 84.0 | 57.7 | -26.3 |
| SD3 | 80.2 | 65.1 | -15.1 | 74.1 | 68.9 | -5.2 | 71.5 | 64.2 | -7.3 | 78.4 | 68.2 | -10.2 | 82.6 | 53.3 | -29.3 |
| SD3.5 | 83.3 | 66.1 | -17.2 | 76.1 | 68.9 | -7.2 | 73.0 | 64.7 | -8.3 | 81.2 | 69.6 | -11.6 | 84.9 | 51.3 | -33.6 |
| AuraFlow2 | 87.2 | 69.1 | -18.1 | 78.5 | 71.3 | -7.2 | 76.9 | 69.1 | -7.8 | 83.2 | 70.4 | -12.8 | 84.4 | 60.2 | -24.2 |
| DALL-E 3 | 86.9 | 70.5 | -16.4 | 78.3 | 70.2 | -8.1 | 77.5 | 67.2 | -10.3 | 80.6 | 67.8 | -12.8 | 88.3 | 60.6 | -27.7 |
| IterComp | 81.9 | 66.6 | -15.3 | 75.1 | 68.4 | -6.7 | 73.1 | 67.6 | -5.5 | 80.7 | 67.3 | -13.4 | 83.1 | 56.1 | -27.0 |
| RPG | 74.5 | 62.6 | -11.9 | 70.9 | 65.0 | -5.9 | 69.4 | 64.6 | -4.8 | 76.6 | 64.8 | -11.8 | 75.5 | 49.5 | -26.0 |
| RRNET | 81.5 | 67.4 | -14.1 | 75.2 | 67.2 | -8.0 | 73.5 | 67.0 | -6.5 | 78.5 | 68.8 | -9.9 | 80.3 | 57.5 | -22.8 |
| R2F | 83.0 | 67.9 | -15.1 | 76.0 | 68.1 | -7.9 | 74.5 | 68.5 | -6.0 | 81.8 | 69.2 | -12.6 | 59.1 | 49.4 | -9.7 |
| ReBind | 82.8 | 71.1 | -11.7 | 75.0 | 69.9 | -5.1 | 80.9 | 75.5 | -5.4 | 78.6 | 69.6 | -9.0 | 77.0 | 62.7 | -14.3 |
| ReBind $\lambda=2$ | 81.7 | 70.6 | -11.1 | 73.1 | 69.2 | -3.9 | 81.7 | 76.4 | -5.3 | 79.3 | 68.8 | -10.5 | 78.4 | 62.9 | -15.5 |
| ReBind $\lambda=5$ | 82.2 | 69.9 | -12.3 | 74.4 | 69.4 | -5.0 | 81.1 | 75.2 | -5.9 | 79.4 | 69.9 | -9.5 | 79.9 | 63.0 | -16.9 |

Table 7: **Evaluation Triplets for Compositional Generalization.** We assess model generalization across four categories of subject–relation–object triplets. ReBind demonstrates improvements on in-domain and novel objects, novel (similar) relations, yet novel relation (OOD) remains challenging. **Training relations** include: *chasing, riding, lifting, pulling, holding, throwing*. **In-domain** triplets consist entirely of components seen during training: *mouse-chasing-cat, horse-riding-astronaut, monkey-lifting-zoo-trainer*. **Novel Relation (Similar)** includes unseen relations with familiar subjects and objects: *mouse-following-cat, scientist-carrying-fireman*. **Novel Relation (OOD)** includes cases where the relation and/or subject/object are unseen: *baby-feeding-woman, baby-kissing-mother*. **Novel Object** evaluates compositional generalization to unseen subjects or objects while keeping the relation fixed: *zebra-chasing-tiger, mouse-riding-cat, ballerina-lifting-coach*. $\beta$ is defined as Unmatch (U) − Match (M). All scores are reported as percentages. ReBind reduces directional bias on in-domain, similar-relation, and novel-object settings, but generalization to semantically distant (OOD) relations remains limited.

| Model | In-Domain | | | Novel Rel. (Similar) | | | Novel Rel. (OOD) | | | Novel Object | | |
|---|---|---|---|---|---|---|---|---|---|---|---|---|
| | M | U | $\beta$ | M | U | $\beta$ | M | U | $\beta$ | M | U | $\beta$ |
| SDXL | 65.0 | 84.5 | 19.5 | 54.7 | 64.2 | 9.5 | 51.7 | 90.0 | 38.3 | 67.0 | 69.6 | 2.6 |
| ReBind (SDXL) | 71.5 | 72.9 | **1.4** | 41.9 | 47.3 | **5.4** | 54.8 | 91.2 | **36.4** | 70.5 | 71.6 | **1.1** |
| SD3 (pretrained) | 64.8 | 79.20 | 14.4 | 51.2 | 51.8 | **0.6** | 60.7 | 94.1 | 33.4 | 54.60 | 56.10 | 1.5 |
| ReBind (SD3) | 77.5 | 86.50 | **9.0** | 55.3 | 62.9 | 7.6 | 53.2 | 81.9 | **28.7** | 54.7 | 55.2 | **0.5** |

during training: *mouse chasing cat*, *horse riding astronaut*, and *monkey lifting zoo trainer*; (2) **Novel Relation (Similar)** includes unseen relations that are semantically close to training relations: *mouse following cat* and *scientist carrying fireman*; (3) **Novel Relation (OOD)** includes completely unseen and unrelated relations: *baby feeding woman* and *baby kissing mother*; and (4) **Novel Object** evaluates generalization to new entities under seen relations, including *zebra chasing tiger*, *mouse riding cat*, and *ballerina lifting coach*.

Table 7 shows that ReBind reduces directional bias across all categories. In the **In-Domain** setting, it reduces $\beta$ from 19.5 to 1.4 (**92% improvement**) and increases matching by 6.1 points. On **Novel Object**, bias drops from 2.6 to 1.1 (**58%**). On **Novel Relation (Similar)** and **OOD**, bias improves by **43%** and **5%**, respectively. These results demonstrate the effectiveness of intermediate supervision in the multi-relation setting. However, improvements on unfamiliar relations remain limited, suggesting the need for future strategies to address this limitation (see Sec. D).

**Does finetuning on intermediate compositions cause unwanted overfitting on their reverse counterpart?** To assess whether training on *directional intermediate compositions proposed by ReBind* causes overfitting, we evaluate both the *intermediates* used during training (e.g., "mouse chasing cheese") and their *role-reversed counterparts* (e.g., "cheese chasing mouse") in Table 9.

Table 8: **Evaluation on COCO .** We evaluate 5k image-text pairs to assess whether finetuning affects general image quality. While FID increases slightly, CLIPScore remains comparable, indicating that image-text alignment is preserved.

| Model | CLIPScore ↑ | FID ↓ |
|---|---|---|
| SDXL | 21.17 | **83.95** |
| ReBind (SDXL) | **21.20** | 90.56 |

Table 9: **Training on intermediates doesn't introduce significant bias against their reverse counterparts.** Comparison of Matching (M), Unmatching (U), and $\beta$ scores across models for Intermediate (e.g., mouse chasing cheese) and Reverse compositions (cheese chasing mouse). All values are shown as percentages. SDXL and DALL-E 3 are based on spatially-aware prompts.

| Model | Intermediate | | | Reverse | | |
|---|---|---|---|---|---|---|
| | M | U | $\beta \downarrow$ | M | U | $\beta \downarrow$ |
| DALL-E 3 | 82.7 | 70.9 | -11.8 | 81.4 | 71.5 | -9.9 |
| SDXL | 65.5 | 57.8 | -7.7 | 62.5 | 61.9 | **-0.6** |
| ReBind (SDXL) | **80.4** | 67.9 | **-12.5** | **69.9** | 74.8 | 4.9 |

ReBind significantly improves generation quality on intermediates, increasing match score by +14.9 and reducing $\beta$ compared to SDXL, confirming that the model has effectively learned the intended directional role bindings.

Importantly, ReBind also improves matching scores on the reversed versions of the intermediate prompts (69.9 vs. 62.5 for SDXL), even though those reversed prompts were not used during training. The modest increase in bias ($\beta = 4.9$) is expected, as these reversed prompts can be unintuitive or rare themselves (e.g., "cheese chasing mouse") and we trained on their opposite counterparts (i.e., intermediates). These small increases are outweighed by the substantial improvements on rare target compositions (see Table 2), confirming that directional intermediates offer a reliable signal for improving compositional generalization.

**Performance on prompts beyond action-based relations**. We further evaluate whether fine-tuning on relation-based intermediates affects the general image quality of the model. To this end, we generate 5,000 images from COCO 2017 validation prompts [Lin et al., 2014, Chen et al., 2015] using the same configurations for both SDXL and ReBind. As shown in Table 8, the CLIPScore [Hessel et al., 2021] remains nearly identical, indicating that image-text alignment is preserved, while FI [Heusel et al., 2017] increases slightly. Overall, these results confirm that our method does not degrade general generation performance and that LoRA fine-tuning helps retain visual fidelity while improving compositional faithfulness.

**Ablation on number of Intermediates (N)**. We tested N=1,3,5,10 for each active/passive composition. On average, performance showed minimal variation (between 6.60-7.90, optimal at 7.90 for N=20).

**Limitations of Prompt Engineering for Rare Compositions.** In Fig. 11, we explore using GPT-4o to create advanced prompts with detailed spatial positions and facial expressions. While these enhanced prompts slightly improve performance in stronger models like DALL-E 3, they actually worsen results for SDXL and IterComp, likely due to their weaker language understanding capabilities. More importantly, even for advanced models, the improvements remain minimal, demonstrating that prompt engineering alone cannot effectively overcome the T2I models.

**ReVersion failed in representing action-based relations.** We evaluated ReVersion [Huang et al., 2024] by fine-tuning it on frequent compositions using prompts like *{subject} <R> {object}*". However, ReVersion performed poorly on both rare and frequent compositions, failing to represent actions like chasing (see Fig. 12). This aligns with prior findings in [Wu et al., 2024b], which show that ReVersion struggles with complex relational reasoning. Methods like ReVersion may encode a rough embedding of the intended relation in <R>, but **it does not mitigate the existing compositional biases in the base T2I model**. Based on this, we exclude ReVersion from our main quantitative

Table 10: **Ablation on number of examples per prompt on all relations.**

| Type | N | VQAScore | | |
|---|---|---|---|---|
| | | **Match** | **Unmatch** | $\beta$ |
| Freq | 1 | 69.0 | 56.1 | -12.9 |
| | 3 | 72.7 | 58.4 | -14.3 |
| | 5 | 79.0 | 62.6 | -16.4 |
| | 10 | 80.0 | 62.0 | -17.7 |
| Rare | 1 | 59.2 | 66.3 | 7.0 |
| | 3 | 58.3 | 66.0 | 7.7 |
| | 5 | 61.7 | 68.3 | 6.6 |
| | 10 | 60.6 | 68.5 | 7.8 |

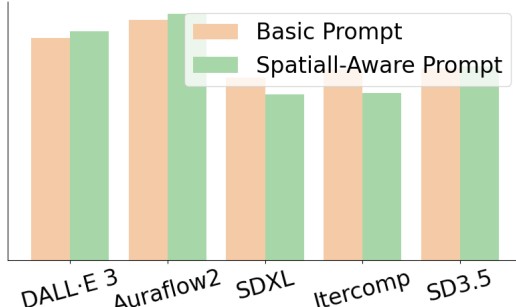

Figure 11: **Spatially-Aware prompt ablation**. Spatially-aware prompts improve strong models but do not fully mitigate counter-stereotypical biases.

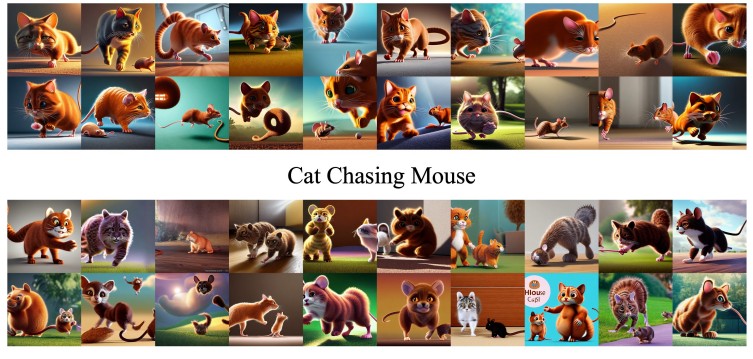

Cat Chasing Mouse

Mouse Chasing Cat

Figure 12: **ReVersion Examples**. ReVersion fails to clearly represent both frequent composition (cat chasing mouse) and rare composition (mouse chasing cat)

comparisons. We also tried a soft-prompt method that learns the entire prompt. It also failed to produce meaningful action generations and required separate training for each relation or prompt.

## D    Limitations

While our framework improves directional role binding and generalization on rare compositions, several limitations remain:

**Multi-Object Interactions.** Fig. 13 illustrates that both pretrained models and our ReBind Models struggle to correctly depict relations involving more than two entities. For instance, the prompt "puppy throwing a ball to a boy" fails to generate coherent spatial and role configurations. This underscores the challenge of extending directional composition frameworks to multi-object action-

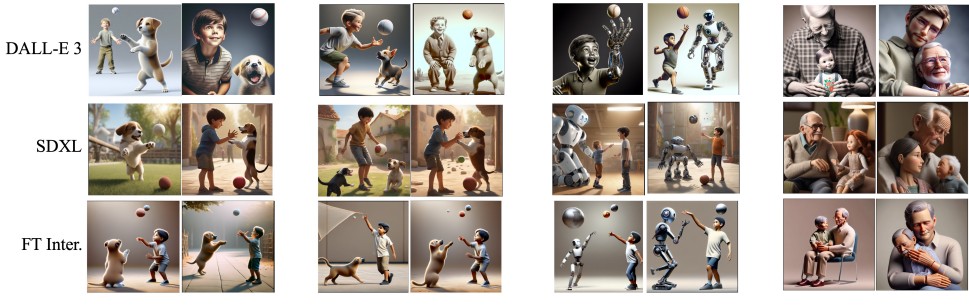

| **Puppy** throwing a ball to a **boy** | **Boy** throwing a ball to a **puppy** | **Boy** throwing a ball to a **robot** | **Doll** holding **Grandpa** |

Figure 13: **Failures**. Baselines and ReBind fail to clearly represent multi-object relations.

based relations. Moreover, we observe that when generating "doll holding grandpa" ReBind and baselines may represent both doll and grandpa as a doll, we conjecture this originates in a lack of context on whether grandpa is a doll or real.

**Visual Artifacts and Image Quality.** We observe that ReBind may introduce visual artifacts, such as object entanglement (e.g., a cat with a mouse tail) and mimic the synthetic training data style. We further evaluated ReBind on COCO in Table 8 and observed comparable CLIPScore and slightly higher FID, suggesting the drop in image quality is not significant. We invite future work to explore strategies for mitigating role collapse in action-based relations without compromising image quality or introducing visual artifacts.

**Generalization limitations.** Table 7 shows that ReBind effectively decomposes rare compositions into learnable intermediate prompts that guide the model toward improved role binding. Our results demonstrate meaningful generalization to novel objects and semantically similar relations. However, generalization to semantically distant or out-of-distribution relations (e.g., "feeding", "kissing") remains limited. Additionally, while intermediate prompts/compositions (e.g., "mouse chasing cheese") help the model internalize directional structure, they may introduce mild bias against their reverse (e.g., "cheese chasing mouse"), particularly when the reverse is itself rare or implausible. This highlights the importance of future extensions to incorporate better training strategies like contrastive learning across relations for preventing such biases.

We emphasize that this is the first work to systematically evaluate role bias in action-based relations, and propose to use active/passive intermediates to improve generalization using a simple approach.

## E   Experimental Settings

Our experiments are organized into two stages:

1. **Benchmarking Pretrained Models and Compositional Methods.** We evaluate a suite of state-of-the-art pre-trained T2I models and compositional generation methods. All models are prompted using either a standard format—"A photo of one {subject} {relation} one {object}"– or a more detailed spatially-aware description using our spatially-aware templates (see Table 11). No model training is involved at this stage.

2. **Testing the Hypothesis of Intermediate Composition.** To evaluate our hypothesis that rare compositions generation can be improved via decomposition into more common intermediate ones, we fine-tune SDXL using LoRA on intermediate compositions selected by our ReBind framework (see Table 5). We evaluate ReBind under two settings: (1) single-relation training and (2) multi-relation training.

**Implementation Details.** For fine-tuning, we use the official HuggingFace LoRA training pipeline[1]. All experiments are conducted on a single NVIDIA L40S GPU for 1000 training steps. We adopt LoRA with rank 32, bfloat16 precision, and use the Adam optimizer with a gradient accumulation

---

[1]https://github.com/huggingface/diffusers/blob/main/examples/text_to_image/train_text_to_image_lora_sdxl.py

Table 11: LLM prompt template for expanding abstract triplet descriptions into detailed, relation-focused image prompts for text-to-image generation.

| **LLM Prompt for Expanding Image Descriptions** |
| --- |
| **System Prompt:** 
 *You are an expert at crafting precise, relation-focused descriptions for text-to-image models. Your specialty is describing the key entities and their interactions in a way that emphasizes their roles, poses, and spatial relationships. Keep descriptions concise yet vivid, using creative visualization when needed (e.g., 'apple with tiny legs running away' to showcase 'mouse chasing cat').* |
| **User Prompt:** 
 *Transform this prompt into a focused description that emphasizes:* 
 1. **Key entities and their states:** 
 • Poses, expressions, and orientations that match their roles 
 • Distinctive features that support the action/relation 
 2. **Spatial positioning** that clearly shows the relationship. Use proper spatial prepositions 
 (e.g., above, below, behind, in front of) to describe the relative position of entities 
 (e.g., 'a mouse is behind a cat running towards the cat' to express 'mouse chasing cat') 
 3. **Simple creative elements** if needed to clarify the interaction 
 4. Images must be **photorealistic**. Include this in the description. 
 5. Ensure **correct entity count** (e.g., 'a mouse is chasing a cat' should have one mouse and one cat) 
 6. Avoid **redundant descriptions** 

 *Input prompt: {prompt}* 

 *Provide a concise, clear description focusing on the entities and their relationship. Avoid background details unless essential to the interaction. Keep it brief but vivid.* |

step size of 4 (with batch size 1 per step). Each intermediate prompt is used to generate 20 training images. A fixed seed (42) is used for all fine-tuning experiments to ensure reproducibility. However, when constructing RoleBench using pretrained models, we do not fix the seed tow promote diversity. For LLM prompting, we use GPT-4o. Our method introduces no inference-time overhead and is computationally lightweight, requiring only LoRA fine-tuning. Across all experiments, average training, inference, and evaluation completes in under 1.5 hours. To ensure training quality, we filter generated training images (more details in Sec. 3 and Sec. 4)

**Training Setup: Single vs. Multi-Relation.** For fair comparison with RRNeT Wu et al. [2024b], we train one model per relation in the main paper's experiments. In Sec . C, Table 7, we evaluated a single ReBind model trained jointly on multiple relations, demonstrating scalability. The prompts for ReBind and evaluation questions are in Sec. F.

# F   Prompts templates & in-context examples

We include prompt templates used for intermediate generation in Table 12, spatially-aware prompt expansion (aka advanced prompt) in Table 11, and Q/A generation in Table 13. Moreover, Table 14 showcases representative in-context examples for binary yes/no question generation while Table 15 demonstrates some in-context examples for active/passive intermediate decomposition.

Table 12: LLM prompt template for generating intermediate active-passive triplets. For each unusual target triplet, the LLM generates easier-to-generate intermediate triplets while maintaining realistic relationships.

---

**LLM Prompt for Intermediate Active-Passive Triplet Generation**

---

**System Prompt:**
*You are an expert at breaking down unusual image prompts into easier-to-generate intermediate steps. Your task is to create multiple distinct intermediate active-passive pairs for each given triplet. Intermediate steps must be realistic.*

---

**User Prompt:**
*Generate intermediate active and passive triplets that are easy for image models to generate.*
**IMPORTANT INSTRUCTIONS:**
1. Each triplet must follow the format `<subject>_<relation>_<object>`.
2. Generate 2-3 passive and active triplets for each target triplet:
   - **Active triplets:** Keep the subject from target triplet, change the object
   - **Passive triplets:** Keep the object from target triplet, change the subject
3. Choose intermediate objects that make the triplet easier to generate:
   - Use objects from different categories (Animal → Human, Human → Object)
   - Prefer frequent/common interactions when possible
   - Ensure physical plausibility (realistic size relationships and actions)
   - Avoid duplicates and ensure diversity in chosen objects

---

**Output Format:**
```
{
  "target_triplet": {
    "passive": [
      {"triplet": "subject1_relation_object"},
      {"triplet": "subject2_relation_object"}
    ],
    "active": [
      {"triplet": "subject_relation_object1"},
      {"triplet": "subject_relation_object2"}
    ]
  }
}
```

---

**Examples:**
*[in-context examples]*

---

**Input Query:**

```
Now generate for:
    input:
        triplet: {target_triplet}
        prompt: {target_prompt}
```

---

Table 13: We generate binary (yes/no) questions per various categories (Spatial, Orientation, Pose, and Facial Expression/Interaction) to obtain fine-grained score.

---

**Evaluation Question Generation Prompt Template**

---

**System Prompt:**
*You are an expert in structured visual relationship assessment. Generate evaluation criteria to systematically verify if images accurately depict specified relational prompts through categorized binary questions.*

---

**User Prompt:**

**Objective:** Generate atomic yes/no questions to evaluate if an image accurately depicts a given prompt describing the image.

**Question Categories:**
1. **Spatial** - Evaluate relative positioning and spatial configuration
2. **Pose** - Assess posture, stance, and motion characteristics
3. **Orientation** - Evaluate directional alignment and facing orientation
4. **Interaction** - Evaluate visual engagement (e.g. gaze, facial expression)

**Key Guidelines:**
1. Questions should evaluate precisely one visual attribute (atomic evaluation)
2. All questions must be formulated to accept binary responses (yes/no)
3. Focus exclusively on objectively observable visual elements
4. Eliminate redundancy and subjective interpretations

---

**Examples:**
*[In-context examples are provided to the model but omitted here for brevity. See Table 14 for complete examples.]*

---

**Input Query:**
Now, generate a similar set of yes/no questions for the following relationship, ensuring each question is atomic and directly observable in an image. Input: prompt

---

Table 14: In-context examples from binary (yes/no) question generation prompt, showing how the model generates structured assessment criteria for different visual relationships.

---

**In-context examples for binary (yes/no) question generation**

---

**Input:** A photo of a cat chasing a mouse
**Output:**
```
{
  "spatial": {
    "questions": [
      "Is the cat positioned behind the mouse, indicating it is chasing? (yes/no)",
      "Is the mouse positioned in front of the cat, indicating it is being chased? (yes/no)"
    ]
  }
} (Additional categories omitted for brevity)
```

---

**Input:** A photo of a horse riding on an astronaut
**Output:**
```
{
  "detection": {
    "questions": [
      "Is there a horse visible in the image? (yes/no)",
      "Is there an astronaut visible in the image? (yes/no)"
    ]
  },
  "spatial": {
    "questions": [
      "Is the horse positioned above the astronaut, suggesting it is riding on them? (yes/no)",
      "Is the astronaut positioned beneath the horse? (yes/no)"
    ]
  }
} (Additional categories omitted for brevity)
```

---

**Input:** A photo of one baby feeding one woman
**Output:**
```
{
  "detection": {
    "questions": [
      "Is there a baby visible in the image? (yes/no)",
      "Is there a woman visible in the image? (yes/no)"
    ]
  },
  "pose": {
    "questions": [
      "Is the baby depicted in a feeding pose, such as extending an arm towards the woman with food? (yes/no)",
      "Is the woman's posture receptive, such as having her mouth open or hands positioned to accept the food? (yes/no)"
    ]
  }
} (Additional categories omitted for brevity)
```

---

Table 15: Representative examples from the Intermediate Active-Passive Triplet Generation prompt showing how the model creates easier-to-generate intermediate triplets by modifying either the subject or object of unusual relation.

---

**In-context examples for the Intermediate Active-Passive Triplet Generation**

**Input:**
triplet: mouse_chasing_cat
prompt: A photo of one mouse chasing one cat
**Output:**
```
{
  "mouse_chasing_cat": {
    "passive": [
      {"triplet": "boy_chasing_cat"},
      {"triplet": "dog_chasing_cat"}
    ],
    "active": [
      {"triplet": "mouse_chasing_boy"},
      {"triplet": "mouse_chasing_cheese"}
    ]
  }
}
```

**Input:**
triplet: horse_riding_astronaut
prompt: A photo of one horse riding one astronaut
**Output:**
```
{
  "horse_riding_astronaut": {
    "passive": [
      {"triplet": "bear_riding_astronaut"},
      {"triplet": "dog_riding_astronaut"}
    ],
    "active": [
      {"triplet": "horse_riding_bear"},
      {"triplet": "horse_riding_dog"}
    ]
  }
}
```

**Input:**
triplet: baby_feeding_woman
prompt: A photo of one baby feeding one woman
**Output:**
```
{
  "baby_feeding_woman": {
    "passive": [
      {"triplet": "robot_feeding_woman"}
    ],
    "active": [
      {"triplet": "baby_feeding_doll"}
    ]
  }
}
```

