# OpenReview forum: "Role Bias in Diffusion Models: Diagnosing and Mitigating through Intermediate Decomposition"
_NeurIPS.cc/2025/Conference — NeurIPS 2025 poster_

### Official Review · Reviewer_jpRP · 2025-06-29

**Clarity:** 3
**Significance:** 4
**Originality:** 4
**Rating:** 5
**Confidence:** 4

**Summary:**

This paper investigates an inherent bias in text-to-image diffusion models and a phenomenon that the authors call role collapse. In particular, relations with reversed order, i.e., from an active role to a passive role, can indicate possible directional biases if one composition is more frequent than another. To address this directional bias, the authors propose a novel framework, namely ReBind, that incorporates intermediate compositions to isolate and bridge two objects with unrelated objects. These intermediates' prompts are generated by LLMs. Experiments demonstrate the existence of role bias in advanced T2I models, and evaluations are widely conducted to verify their fine-tuned models with improved compositional generative ability.

**Questions:**

1. What's the criterion to determine "frequent"/"rare" relations? In Appendix 2, the authors mention that frequent compositions follow typical role priors. Is this prior manually annotated or statistically analyzed from the training dataset?
2. I found the analysis of role bias is conducted by analyzing generated images of T2I models, i.e., annotating Match/Unmatch ratio in Table 1. Could the authors provide a statistical analysis of the training data? For example, the possibility of each object to be an active role v.s. a passive role in prompts. This might help to convey the concept in a more straightforward way and to complement your statement in Sec. 3.2. Also, adding links to related figures/tables after these statements may help the readers to follow.
3. Is it possible to analyze the intermediate features to indicate the role collapse issue? Take the rare composition "mouse chasing cat" as an example, if cross-attention visualization indicates overlap areas between the relation word "chasing" and the passive subject "cat" rather than the ideal object "mouse", this might be a sign of incorrect role binding.
4. Although I appreciate that the authors highlight terms in italic, I personally think some definitions are not clear. For example, in line 187, the authors term "mouse" and "dog" as "cross-categorical objects". I think a detailed classification like "animal / plant / human / non-living" will clarify this, i.e., "mouse" and "dog" are both animals, while "boy"/"cheese" belong human/non-living are "cross-categorical objects".
5. The influence of the hyperparameter N is not studied.

Minor suggestions:
1. To explain the role collapse phenomenon, the authors claim that previous works attribute the issue to "the lack of rare samples and role diversity in training data," while this paper suggests "the overwhelming presence of their frequent counterparts" (e.g., lines 51-53). In my opinion, the difference between "rare" (previous) and "biased distribution" (yours) is somewhat vague. While I agree that this paper has conducted in-depth investigations and comprehensive analyses than prior studies. I recommend that the authors rephrase these sentences.
2. Fig. 1 (b) "Mouse chasing cat" shows the issue of object entanglement, i.e., the "cat" has a mouse-like hands, feet, and tail. I recommend that the authors discuss this  (potentially due to the inherent entanglement when generating multiple objects) in the section of limitations.
3. Should Appendix Fig. 1 w be $\lambda$?
4. Replace the name "FT Inter" with the method name "ReBind" in the result tables/figures.
5. The following literatures have similar focuses to this paper that investigates attribute-based bias or quantifies the bias for the authors' consideration to inspire your future work.

[1] Magnet: We never know how text-to-image diffusion models work, until we learn how vision-language models function.

[2] Openbias: Open-set bias detection in text-toimage generative models.

**Ethical Concerns:**

["NO or VERY MINOR ethics concerns only"]

**Final Justification:**

This paper investigates action-based relations and provides a comprehensive and systematic analysis of frequent / rare relations. The authors identify such a bias inherent in the training datasets that affects various T2I models. Motivated by this, the authors propose ReBind that fine-tunes T2I models using the LoRA technique. Experiments validate the method improves compositional generation both quantitatively and qualitatively across five pretrained text-to-image models. While some statement lack clarity particularly in the analysis, I believe these can be addressed through careful revisions. I therefore recommend a score of accept.

**Limitations:**

The authors discuss the limitations of this work in Appendix Sec. 4.

**Quality:**

3

**Strengths And Weaknesses:**

**Strengths**: This paper indicates a critical relation bias in the text-to-image diffusion models that hinders faithful compositional generation. The authors provide quantitative analysis on various T2I models to confirm their hypothesis. Next, a simple yet efficient approach is proposed that decouples original rare relations into intermediate frequent compositions with LLM-generated high-quality intermediates to fine-tune the T2I model using LoRA. Experiments are widely conducted with appropriate metrics.

**Weaknesses**: Although the experiment shows that existing T2I models present role bias, a fundamental statistical analysis of the training dataset solely to exhibit such directional bias is missing. Also, some descriptions are overstated (when compared with previous literature) or lack clarity (these terms in italic, such as "directional bias", "cross-categorical objects"). Some relations are not studied, e.g., spatial ("A in the B"), or not well addressed, e.g., rare actions ("A is kissing B").

---

> ### Author Rebuttal · Authors · 2025-07-31
>
> We are extremely grateful for the extremely comprehensive review. We appreciate the positive comments, particularly those considering our **experiments and analysis comprehensive and in-depth**, as well as **considering our work to indicate a critical bias** in T2I models. We appreciate reviewer jpRP for their thoughtful and comprehensive feedback regarding improving the quality of the current manuscript and our future works! We address specific concerns and questions.
>
> > [W1] ... experiment shows existing T2I models present role bias, ... statistical analysis of the training dataset solely to exhibit such directional bias is missing.
>
> We appreciate the reviewer's feedback on training data analysis and agree that data analysis could be interesting and insightful about T2I models. Unfortunately, **recent T2I models use closed-source training data, making direct statistical analysis infeasible. We therefore employ established proxy methods validated in recent literature: (1) semantic plausibility (e.g., knowing that "cat chasing mouse" is more plausible than "mouse chasing cat") analysis using LLM logprobs and (2) Google Search frequency counts.** The results show similar patterns to our T2I benchmarking results (i.e., frequent compositions have higher semantic plausibility and higher frequency on Google Search) -- **We'll include an in-depth discussion in the supplement.**
>
>
> **Since LLMs and T2I models are trained on similar web data, we assume their data follows similar directional bias patterns**. Following [2], we use LLM (i.e., Mistral-7B) logprobs as a proxy for training data through semantic plausibility estimation. Semantic plausibility effectively captures world knowledge and frequency patterns [3], allowing us to estimate relative frequency in training data (higher plausibility indicates higher frequency).
>
>
> **Our analysis shows frequent compositions achieve higher plausibility in 80% of cases, indicating similar directional bias in the underlying training data.** Google Search counts confirm this pattern with an average rare-to-frequent ratio of 0.21 (e.g., count("mouse chasing cat") / count("cat chasing mouse")). **These results align with our T2I benchmark evaluation, where models performed better on frequent compositions compared to rare compositions.**
>
>
>
> > Some relations are not studied, e.g., spatial ("A in the B"), or not well addressed, e.g., rare actions ("A is kissing B").
>
>
> We thank the reviewer for this constructive feedback and **will include quantitative and qualitative results on each action (e.g., kissing) in the supplement**.
>
> We agree that studying role bias across diverse relation types could strengthen this work. **We focus on action-based relations because they naturally exhibit directional role bias.** Many prior works have studied spatial relations, showing compositional generation methods (e.g., bbox and LLM-based approaches) are effective (see related work section). Table 2 demonstrates these methods are not as effective on action-based relations because correctly representing actions requires beyond spatial understanding (e.g., positioning the mouse behind the cat is not enough to show chasing; correct orientation, pose, and facial expression are needed)
>
> We see this work as providing insights on role-directional bias in action-based compositions, and we look forward to more robust work that can generalize across all relation types.
>
>
> > [Q1] What's the criterion to determine "frequent"/"rare" relations? ... frequent compositions follow typical role priors. Is this prior manually annotated or statistically analyzed from the training dataset?
>
> It's prior annotated based on semantic plausibility. A more typical action (e.g., cat chasing mouse) is more likely. Hence, it's more frequent.
>
>
> > [Q2] ...  Also, adding links to related figures/tables after these statements may help the readers to follow.
>
> Thanks for the suggestion! We will revise Sec. 3.2 and proofread others to provide links to relevant figures/tables.
>
>
> > [Q3] Is it possible to analyze the intermediate features to indicate the role collapse issue? .... cross-attention visualization ....
>
> We appreciate the reviewer for their great suggestion. Due to limited time, we manually checked visualized cross-attention weights and will include in the supplement (cannot upload images in the OpenReview). We observed more issues on rare compositions such as overlap between objects and overlap between verb (e.g., chasing) and passive role (i.e. cat) in mouse chasing cat. We'll explore more in the future for both diagnosis and bias mitigation.
>
>
>
>
> >[Q5] The influence of the hyperparameter N is not studied.
>
> We tested N=1,3,5,10 for each active/passive composition. On average, performance showed minimal variation ($\beta$ between 6.60-7.90, optimal at 7.90 for N=20). We will include complete results in the supplement. This shows that our active/passive intermediate finetuning can be effective even without requiring a significant amount of data. We further explored multi-relation (i.e., higher number of examples and more diversity) in Table 4 in supp which shows promising results on generalizing to novel objects and relations.
>
>
> > ....some descriptions are overstated.... or lack clarity (these terms in italic, such as "directional bias", "cross-categorical objects").
>
>
> Thanks for the constructive feedback! We will proofread, read modify the text to enhance clarity.
> We updated L187 to clarify cross-categorical selection: We observe that selecting objects (i.e., agent and patient roles) from different categories (e.g., human, animals, or non-living objects) can be effective to satisfy this property. For instance, "mouse chasing dog" (both agent and patient are animals) is a poor choice because "dog chasing mouse" is more likely to be frequent. On the other hand, "mouse chasing boy/cheese" (agent is animal, patient is human/non-living object) is more appropriate as the reverse counterpart ("boy/cheese chasing mouse") is less likely to be frequent.
>
>
> ## Minor Suggestion
> We greatly appreciate the reviewer for providing valuable suggestions. We have addressed the typos and improved the clarity of the text per suggestions.
>
> * Thanks, We will rephrase to clarify: prior work studies low frequency of rare compositions, while our unique focus is on how high frequency of reverse compositions specifically impedes rare generation, which can be addressed through intermediates.
>
> * As suggested, we added a discussion on image quality and artifacts to our Limitation section. We also included the new experiment on COCO-Caption results (see W2 for response to YW6c)
>
> * yes, we updated w -> $\lambda$ and  FT. Inter -> ReBind (in tables, figures and text)
>
> * We thank the reviewer for these valuable references on quantifying bias and attribute-based bias, which offer complementary perspectives to our focus on action-based compositional bias. We will add them to our Related Work section and our future works.
>
>
>
> [1] Log Probabilities Are a Reliable Estimate of Semantic Plausibility in Base and Instruction-Tuned Language Models, ACL 2024
>
> [2] Did the Cat Drink the Coffee? Challenging Transformers with Generalized Event Knowledge, ACL 2021
>
> [3] Impact of Co-occurrence on Factual Knowledge of Large Language Models, EMNLP 2023
>
>
> We hope that we have addressed the existing concerns and are happy to discuss more about them or if any other concerns exist.

---

> > ### Comment · Reviewer_jpRP · 2025-08-01
> > **Response to rebuttal**
> >
> > Thank you for the explanations! It is good to see that the authors engaged with all the questions, for me especially their efforts to provide a statistical analysis in two alternative ways. I believe my score is appropriate, and I look forward to seeing your future exploration!

---

> > > ### Author Response · Authors · 2025-08-01
> > >
> > > Thank you for your kind response!

---

### Official Review · Reviewer_9JWJ · 2025-07-01

**Clarity:** 3
**Significance:** 2
**Originality:** 3
**Rating:** 4
**Confidence:** 4

**Summary:**

This paper investigates common failures of Text-to-Image (T2I) diffusion models in generating rare compositions of subjects and actions (e.g., a mouse chasing a cat), which the authors refer to as "Role-Collapse." To address this issue, the authors introduce a new benchmark, RoleBench, designed to evaluate compositional generalization specifically for action-subject pairs. An empirical study using this benchmark reveals that T2I models often fail to generate rare compositions when the training dataset exhibits strong directional biases regarding subject-action relationships (e.g., cats usually chase mice, but mice rarely chase cats). Motivated by this observation, the authors propose mitigating these directional biases by fine-tuning T2I models with intermediate examples that replace active/passive subjects with ones having low directional biases. Experimental results indicate that fine-tuning on these intermediate examples indeed reduces role biases. The main contributions of the paper are as follows:
- Identification of failures in compositional generalization for action-subject pairs in T2I models, attributed primarily to strong directional biases present in the training dataset.
- Introduction of a new benchmark, RoleBench, to systematically examine directional biases in compositional generalization related to action-based relationships.
- Proposal of a simple yet effective approach to mitigate directional biases by decomposing rare action-subject relationships into intermediate action-subject pairs that have low directional biases.

**Questions:**

- How did the authors determine the frequent and rare cases when constructing RoleBench? Were these cases manually identified based on common sense or intuition? If so, how can we ensure that such cases are indeed frequent/rare or counterparts of intermediates are indeed rare?

- L232-234 states that “Results show that FT-Inter generalizes to rare compositions without overfitting, while Freq/Rare FT overfits to frequent compositions”. However, it is not clear which results it is referencing (maybe Table 3?) and how such results are evidence of overfitting.

- (Minor suggestion) Providing empirical evidence of L156-158 (maybe in appendix) would improve the completeness and soundness of the author's claim.

- (Minor suggestion) It would be more clear if the abbreviation of FT Int. is clearly stated (e.g., directly connected to the main method).

- (Minor suggestion) Last column of Figure 5 seems wrongly inserted (maybe both image and caption..?) in Supplementary material.

**Ethical Concerns:**

["NO or VERY MINOR ethics concerns only"]

**Final Justification:**

This paper identifies and provides valuable insights into the failures of compositional generalization for action-subject pairs in T2I models, primarily due to strong directional biases in the training dataset. In addition to introducing a new benchmark, RoleBench, to systematically examine directional biases in action-based relationships, the authors propose a simple yet effective approach to mitigate such biases by decomposing rare action-subject relationships into intermediate action-subject pairs with lower directional biases.
A clear limitation of the approach is that it requires some manual inspection to construct freq/rare pairs, which may limit the scalability of this work. Despite this limitation, I believe this approach provides valuable insights into the underexplored failure mode (role bias) in the compositional generalization of T2I diffusion models and the authors effectively address this issue within a manageable scope. Therefore, I am leaning towards acceptance.

**Limitations:**

yes

**Paper Formatting Concerns:**

I did not find any major formatting issues.

**Quality:**

2

**Strengths And Weaknesses:**

**Strengths**
- The paper is well-written and easy to follow.
- It addresses an underexplored failure mode (role bias) in compositional generalization of T2I diffusion models.
- The idea of role decomposition and finetuning with intermediate examples containing uncommon role bindings (e.g., mouse as chaser and cat as target) is novel and sound. The empirical results also demonstrate that this straightforward approach is indeed effective.

**Weaknesses**
- While the authors claim that the common failures of T2I diffusion models in generating correct visual relationships between objects are due to the presence of frequent counterparts rather than the absence of rare compositions in the training dataset (e.g., L8–12 in the abstract, L52–53 in the introduction), it seems it lacks direct evidence to support this claim. To clearly validate this claim, the authors should directly compare the bias ($\beta$) of rare compositions that have frequent counterparts (e.g., mouse chasing cat) with rare compositions that do not have such counterparts (e.g., horse riding astronaut vs. horse riding bear). Although RoleBench constructs Active/Passive Intermediates guided by this principle, and results show improvement, explicitly demonstrating that the bias ($\beta$) for Active/Passive Intermediates is indeed smaller than the original examples would better support the claim.
- Although the examples in RoleBench (listed in Table 1 in the appendix) appear reasonable, each case seems to be manually determined. The authors did not use quantitative metrics to validate whether selected cases are indeed frequent or rare. Therefore, it's unclear whether the selected cases truly represent frequent instances, or whether their counterparts (active/passive intermediates) are actually rare, weakening the authors' overall claim.

---

> ### Author Rebuttal · Authors · 2025-07-31
>
> We thank Reviewer 9JWJ for their review. We appreciate the positive comments, particularly those about the novelty and soundness of the idea and the effectiveness of the method. We also thank them for their constructive feedback and for addressing each specific concern
>
> > [W1] ... failures .... are due to the presence of frequent counterparts rather than the absence of rare compositions ... To clearly validate this claim, ... directly compare the bias ... of rare compositions that have frequent counterparts (e.g., mouse chasing cat) with rare compositions that do not have such counterparts (e.g., horse riding astronaut vs. horse riding bear). ... Explicitly demonstrating that the bias ... for Active/Passive Intermediates is indeed smaller than the original examples would better support the claim.
>
> Thank you for this insightful question. **We agree that comparing the bias $\beta$ of intermediate compositions to rare compositions validates this hypothesis.** We kindly refer to Table 4 (supplement) for this comparison -- **given the extra page, we'll move this table and discussion to the main paper**.
>
>
> **Intermediates achieve negative $\beta$ values while rare compositions show large positive $\beta$, directly supporting our claim.** Intermediates (e.g., "mouse chasing boy") achieve negative $\beta$ (DALL-E 3: -11.8, SDXL: -8.7), indicating successful generation without defaulting to its reverse composition (Negative $\beta$ indicates no bias towards reversed prompt; see L254). This contrasts sharply with rare compositions (e.g., "mouse chasing cat") in Table 1, which show large positive $\beta$ (SDXL: 16.00, DALL-E 3: 23.10), demonstrating bias toward frequent prompt/composition.
>
>
> Further, Table 1 shows $\beta$ is largely positive for rare composition but negative for frequent ones, indicating high bias toward frequent patterns in both cases. However, Table 4 shows a different trend; $\beta$ is negative for both intermediates and their reverses, with smaller magnitudes (e.g., $\beta = +16.00/-27.70$ for rare/freq vs. $\beta = -11.8/-9.9$ for intermediate/reversed in DALL-E 3). **This shows ReBind effectively creates intermediates that generate well in both directions, avoiding the role-bias problem as in rare/freq**.
>
>
>
> > [W2] ... RoleBench ... appear reasonable, each case seems to be manually determined. ... did not use quantitative metrics to validate whether selected cases are indeed frequent or rare. Therefore, it's unclear whether the selected cases truly represent frequent instances or whether their counterparts (active/passive intermediates) are actually rare, weakening the authors' overall claim.
>
> We thank the reviewer for this insightful feedback about our frequent/rare selection and benchmark validation. **We provide elaboration below and include a detailed discussion in supplement.**
>
> **Since the training data of recent T2I models is not publicly available, we adopt semantic plausibility as a proxy for frequency patterns.** Semantic plausibility (knowing that "cat chasing mouse" is more plausible than "mouse chasing cat") effectively captures world knowledge and correlates with training data frequency [3,4,5]. **Since LLMs and T2I models share similar web-based training data, LLM plausibility estimates provide reliable proxies for T2I model biases.**
>
>
> > [Q1.a] How did the authors determine the frequent and rare cases when constructing RoleBench? Were these cases manually identified based on common sense or intuition?
>
>
> First, we manually selected common actions from [1,2] that represent visually distinguishable animate-animate interactions (e.g., "A chasing B" vs "B chasing A" is clear, while "A fighting B" vs "B fighting A" is not). We then used a hybrid approach: GPT-4o generates freq/rare candidate pairs, followed by manual filtering.
>
> **For freq/rare selection:** We restrict to animated objects because inanimate objects create impossible relations [5] (e.g., "chair kissing person"). We created candidate pairs by combining related works data [1] and prompting GPT-4o to generate additional pairs where one direction is plausible (frequent) and the reverse is implausible (rare). Annotators manually filter to ensure proper structure. We select one pair per relation ensuring object diversity (e.g., not using mouse/cat for all relations).
>
>
> > [Q1. b] If so, how can we ensure that such cases are indeed frequent/rare or counterparts of intermediates are indeed rare?
>
> We kindly refer to Reviewer jpRP answer on [W1] for details. We briefly discuss:
>
> **We employ LLM logprobs and Google Search counts to validate our freq/rare selection**. Results show frequent compositions achieve higher plausibility in 80% of cases, while Google Search reveals an average rare-to-frequent ratio of 0.21. This, along with our comprehensive T2I evaluation in Table 1 confirms the validity of our rare/frequent compositions.
>
> **Intermediates are not selected manually.** ReBind automatically selects plausible intermediates (e.g., mouse chasing boy) by choosing objects from different categories (e.g., human/animal; see L178-L186) to ensure the reverse is not common. Table 4 (in supplementary) shows intermediates achieve negative $\beta$ values, quantitatively confirming that generated images are more aligned with the intended composition (i.e., mouse chasing boy) and generatable. As described in L197, we further use $\beta <0$ to filter low-quality intermediates.
>
> **Hence, this validates both rare/freq selection in RoleBench and the validity of active/passive intermediates.** We see this work as the first to provide insights on role bias in action-based relations, **rather than proposing an optimal method.** We look forward to future works, building on this direction to develop more robust and cleaner extraction methods.
>
>
> > [Q2] ... “Results show that FT-Inter generalizes to rare compositions without overfitting, while Freq/Rare FT overfits to frequent compositions”. ... which results it is referencing (maybe Table 3?) and how such results are evidence of overfitting.
>
> We thank the reviewer for bringing up this point. Results refer to **Table 3, Figure 3 (human eval), and qualitative examples in Figure 5-main and Figure 1-supp, which show direct finetuning on Freq/Rare is ineffective compared to intermediate finetuning**. We rename "overfitting" to "ineffectiveness" to avoid confusion.
>
> **Human evaluation shows annotators preferred FT Inter over FT Freq/Rare by 71%**, while **Table 3 shows $\beta = 7.9$, $14.7$ for FT Inter and FT Freq/Rare, respectively**. Qualitative results demonstrate **FT Freq/Rare defaults to frequent compositions** ("cat chasing mouse"); e.g., see Fig1 and Fig2 in supplement. This shows a failure to learn rare compositions despite training directly on them. This provides evidence that T2I models cannot effectively generate rare compositions and default to frequent counterparts.
>
> ### Minor Suggestions
> We thank the reviewer for all the constructive and valuable suggestions. We briefly address below:
>
> * We further clarified compositional generation baselines (Table 2) in the text (Sec. 4-Baselines). Briefly: RPG uses region-aware LLM planning, SLD refines images using layout-aware feedback using LLM and object detector; yet they are ineffective, especially on non-spatial criteria (e.g., facial expression).
> * Renamed FT. Inter to ReBind, and last col of Fig 5 to "doll holding grandpa"
>
>
> [1] Relation Rectification in Diffusion Model, CVPR 2024
>
> [2] Verbs Matter: A Tutorial for Determining Verb Difficulty, American journal of speech-language pathology, 2023
>
> [3] Did the Cat Drink the Coffee? Challenging Transformers with Generalized Event Knowledge, ACL 2021
>
> [4] Log Probabilities Are a Reliable Estimate of Semantic Plausibility in Base and Instruction-Tuned Language Models, ACL 2024
>
> [5] Event knowledge in large language models: the gap between the impossible and the unlikely
>
>
> We hope that we have addressed the existing concerns and are happy to discuss more about them or if any other concerns exist.

---

> > ### Author Response · Authors · 2025-08-05
> >
> > Dear Reviewer 9JWJ, thank you again for your time and constructive feedback. We hope that our additional quantitative analysis on the benchmark as well as our detailed response and clarificaiton especially regarding comparison of intermediates vs. rare compositions has helped to address your questions. We’d like to kindly follow up and ask whether the questions are addressed. If you have further questions, we would be happy to discuss.

---

> > > ### Comment · Reviewer_9JWJ · 2025-08-06
> > >
> > > My primary concern was regarding the authors' approach in selecting and validating frequent/rare relations. The results from the LLM logprobs and Google Search counts appear to support the actual frequencies of these frequent/rare relations. While it seems that some manual inspection is still required (e.g., manually selecting common actions that represent visually distinguishable animate-animate interactions or manually filtering frequent/rare pairs after GPT-4o generation), this approach is reasonable at the current stage of the work. It provides valuable insights into role bias in action-based relations and presents empirical studies as proof-of-concept. Therefore, I would like to maintain my score.

---

### Official Review · Reviewer_Tzxi · 2025-07-01

**Clarity:** 4
**Significance:** 3
**Originality:** 3
**Rating:** 4
**Confidence:** 4

**Summary:**

The paper tries to investigate role bias in text to image diffusion models, where models consistently default to generating frequent, stereotypical subject-object relationships. They further introduce a benchmark to evaluate such failure modes where the models don't follow the prompt exactly, but just use the words present and default to the most frequent arrangement of those words. They introduce a framework that decomposes prompts into intermediates which are more learnable.

**Questions:**

1. Can the method be expanded to more complex object action relationships? If so what would be the way to adapt it for more and more such relations?
2. How much improvement can be seed for other image generation models with your method?
3. Can you expand more on the possibility of systematic failures if intermediate choices are suboptimal?

**Ethical Concerns:**

["NO or VERY MINOR ethics concerns only"]

**Final Justification:**

The authors addressed majority of the concerns I had, so I increased my score and am inclined towards acceptance. The paper gives detailed insights into the issue and can help further works to address it even better.

**Limitations:**

Yes

**Quality:**

2

**Strengths And Weaknesses:**

Strengths:
1. The paper is well and clearly written.
2. The problem they try to solve has high significance and they have shown that some of the latest models do have those issues.
3. They showed the effect on performance on the regular prompts after using their method, which is an important ablation.


Weakness:
1. Although the authors show improvement in the given experiments, the benchmark is quite small and mainly involves animals and humans. Broader real world diversity, multi-object action relationships, more hierarchical relationships, etc.  could help strengthen the contribution and evaluation of the models.
2. Although they show that latest models have the bias, their improvement is not shown on them. Only SDXL seems to be used to show the improvement. Showing improvement for other models might help understand the generalization capabilities of the proposed method.

---

> ### Author Rebuttal · Authors · 2025-07-31
>
> We thank reviewer Tzxi for their constructive feedback, and we are particularly glad they found the **problem significant** and ablations important and effective. We respond to specific concerns and questions.
>
>
> > [W1] Although the authors show improvement in the given experiments, the benchmark is quite small and mainly involves animals and humans. Broader real-world diversity, multi-object action relationships, more hierarchical relationships, etc. could help strengthen the contribution and evaluation of the models.
>
> We appreciate the thoughtful comment.
>
> **Our benchmark serves as a controlled study of directional bias effects in action verbs (i.e., does generation of rare composition default to the frequent patterns?)** We intentionally focus on animate-animate interactions where **both directions remain possible but have different plausibility** (cat chasing mouse vs. mouse chasing cat). **This setup isolates compositional bias from impossibility effects.**
>
>
> We elaborate in 3 parts:
>
> * **Multi-object relations.** We ask the same question, and **already discussed this in our limitation (Section 4, supplement; L132)**. We studied a multi-object example ("boy throwing ball to puppy") in our benchmark. Even though initial results showed improvement ($\beta$ reduced by 0.17 points), all T2I models struggled with multi-object generation regardless of direction (Fig. 5, supplement). This makes it difficult to isolate role bias from scene complexity.
>
>
>
> * **Human/Animal Interactions** Including inanimate objects (e.g., inanimate-animate: "backpack lifting person" and inanimate-inanimate: "cloud chasing sun") **makes these relations impossible rather than implausible** [1,2]. Therefore, failure may stem from impossibility rather than role bias (i.e. model is not able to generate "sun chasing cloud" or "backpack lifting person" regardless of whether the reverse composition is frequent -- **we'll clarify in Sec. 4, Data**
>
>
> * **Data size**. RoleBench covers 10 action verbs, 60+ compositions across rare/frequent/intermediate cases, 120+ total prompts, evaluated on 5 SOTA models and 5 compositional methods. **This targeted approach reveals consistent directional bias patterns underexplored in large-scale benchmarks.** We see this work as a controlled study on action-based relations and role biases rather than providing another comprehensive T2I benchmark.
>
>
>
> [1] Event knowledge in large language models: the gap between the impossible and the unlikely. Cognitive Science,2023
> [2] Log Probabilities Are a Reliable Estimate of Semantic Plausibility in Base and Instruction-Tuned Language Models, ACL 2024
>
> > [Q1] Can the method be expanded to more complex object action relationships? If so what would be the way to adapt it for more and more such relations?
>
> We kindly refer to W1 for detailed discussion. Briefly:
>
> **Our framework could extend to complex multi-object relations by generating active/passive intermediates** (e.g., keeping the agent fixed while changing patient roles). While our initial results multi-object example ("dog throwing ball to a puppy") show a reduction in bias. We further observe that **SOTA T2I models struggle with faithful multi-object generation and representing the action (regardless of direction). This makes reliable evaluation challenging as failures could stem from role bias or scene/action complexity.** Hence, we focus mostly on binary relations to establish clear evidence of directional bias effects before tackling more complex scenarios.
>
>
>
> > [W2] Although they show that the latest models have the bias, their improvement is not shown in them. ..... Showing improvement for other models might help understand the generalization capabilities of the proposed method.
>
> Due to limited time, we briefly trained SD3-medium on the chasing relation. We observe a similar pattern: $\beta$ drops by 4.2 points from 11.2 to 7.0 compared to SD3 baseline, and the Matching score increases by 52.7 (from 33.5 to 86.2), showing both **reduced bias and better alignment**. Qualitative outputs indicate significantly fewer artifacts (e.g., object entanglement) than SDXL due to the stronger backbone (results will be included in the supplement as we can't include images in OpenReview). Showcasing improved generalizability on rare composition across two of the most popular models.
>
>
> We highlight that our framework finetunes T2I model on proposed active/passive examples with standard supervised training, making the training easily adaptable and results generalizable as no architectural changes are made. Hence, we believe that this is a minor limitation.
>
> > [Q2] How much improvement can be seen for other image generation models with your method?
>
> We kindly ask that to refer to W2 for more details. We briefly discuss:
>
>
> **We observed similar promising trends on SD3: $\beta$ decreased while matching scores increased, showing both reduced bias and better alignment**. Given these results and our framework's simplicity, we expect similar behavior across other architectures. **SDXL's superior performance over SOTA T2I models (Fig. 3, Table 2) and compositional methods that use LLM planning, image refinement with bboxes, and DPO training further supports this expectation.**
>
>
> > [Q3] Can you expand more on the possibility of systematic failures if intermediate choices are suboptimal?
>
>
> **ReBind filters intermediates to prevent systematic failures**. Concretely, it generates multiple active/passive candidates, creates several images for each, then selects plausible training samples using $\beta$ < 0 to ensure alignment with intended intermediates rather than reversed directions (L197-199).
>
>
> Furthermore, ReBind chooses objects that create more plausible intermediate compositions (L178), which are significantly more plausible than rare compositions and avoid role collapse. Table 4 shows that both DALL-E 3 and SDXL achieve negative β values (-11.8 and -8.7) for intermediates, while Table 1 shows positive $\beta$ values for rare compositions (12.10 for SDXL, 9.20 for DALL-E 3).
>
>
>
> We observed that some visual artifacts persist from intermediate to final outputs, and if intermediates themselves are rare or non-generatable (e.g., inanimate-animate or multi-object relations), we expect this to impact generalization to some degree. However, we see this work as the first to **provide insights on role-directional bias in action-based relations and demonstrate improvement through active/passive intermediates, rather than proposing an optimal model**.
>
>
>
> We hope that we have addressed the existing concerns and are happy to discuss more about them or if any other concerns exist.

---

### Official Review · Reviewer_YW6c · 2025-07-03

**Clarity:** 2
**Significance:** 3
**Originality:** 3
**Rating:** 4
**Confidence:** 5

**Summary:**

The paper proposes a less addressed setup in compositional image generation, i.e. role binding, where certain relationship such as "chasing" appear more frequent for certain order of object and subjects, i.e. cat chasing mouse. They introduced a benchmark dataset of 10 common active relationships with this property, and evaluated any model outputs on such prompts with a modified VQAScore customized to be more accurate by rephrasing the spatial assertions implied by the prompt. They observe that most models fail to get the order of object and subject right in the generated image, and importantly this happens more so in prompts that their reversed object/subject order is extremely more frequent. In addition, when both orders of subj./obj. are rare, the model could bind the roles more accurately. This means that such role binding failure is more attributed to the *imbalanced* frequency of subj./obj. orders in the real world. As a result, they also introduced synthetic data generation in which they decompose triplets of such relations into intermediate triplets that are more easier for T2I models to generate using LLMs and DALL-E3 models. They fine tune (through LoRA) the T2I model using the mentioned dataset and observe better alignment compared to other models in various compositional challenges (table 2).

**Questions:**

See above. I am open to increase my score should the authors address concerns.

**Ethical Concerns:**

["NO or VERY MINOR ethics concerns only"]

**Final Justification:**

Authors addressed my major concerns.

**Quality:**

3

**Strengths And Weaknesses:**

The idea of the paper sounds interesting. I have these concerns:

- What relations were used for the training and fine-tuning? Do they overlap with those of the training relationships?
- As the fine-tuning is done only on specific "relation-based" prompts, one may expect the model to lose its performance on other prompts not concerning subj./relation/obj. and/or on other criteria such as image quality. However, such evaluations seem to be missing in the paper.
- Looking at the last column of table 2, VQAScore, it seems like that although the gap between the unmatched and matched cases is reduced in the right direction, both matched and unmatched scores reduced in the proposed method. That is, it seems that the bias is decreased but it did not necessarily made the model more accurate.
- The authors mentioned in line 181 that "We conjecture that these intermediates separately reinforce uncommon role bindings (e.g., mouse as chaser, cat as target), enabling the model to reconstruct the full rare relation more faithfully during generation." This conjecture was not theoretically or empirically discussed and little insights were given to support this claim.

---

> ### Author Rebuttal · Authors · 2025-07-31
>
> We appreciate Reviewer YW6c for their thoughtful review and are happy that they found this paper interesting. We address each specific concern inline.
>
> >[W1] What relations were used for the training and fine-tuning? Do they overlap with those of the training relationships?
>
>
> **We only fine-tune pre-trained models.** **Yes, all relations overlap with pre-training data.** We intentionally study common actions (relations): chasing, riding, feeding, kissing, throwing, lifting, carrying, following, pulling, and holding that the model has likely encountered during pre-training on large-scale web data (see Table 1 in supp).
>
>
> **Table 1 confirms models know these relations**. They can generate Frequent composition (e.g, cat chasing mouse) but fail in generating rare reversal (i.e., mouse chasing cat), collapsing to frequent composition. **The issue isn't action overlap, but it's a rare/frequent imbalance** within the same relation. We study reversal pairs where one direction is common, and the reverse is rare.
>
> **Finetuning settings**. **We fine-tune only on active/passive intermediates** (e.g., "mouse chasing boy"). The model never sees test compositions (i.e., rare/freq). We explore two settings:
>
> * **Per-Relation (Table 2; supp)**: Like [1], we train and test on the same relation.
> *  **Multi-Relation (Table 4; supp)**: train on 6 relations, test on 10 relations (6 seen + 4 unseen).
>
> Results show a significant reduction in $\beta$ with promising generalization across both Per-Relation and Multi-Relation.
>
>
>
> [1] Relation Rectification in Diffusion Model, CVPR 2024
>
> >[W2] ... one may expect the model to lose its performance on other prompts not concerning subj./relation/obj. and/or on other criteria such as image quality. However, such evaluations seem to be missing in the paper.
>
> We acknowledge this potential trade-off and highlight this in limitation supp. Sec. 4. While it would be ideal for there to be no degradation, we find that our method achieves comparable image-text alignment (i.e. CLIPScore) but slightly higher FID on other types of prompts.
>
>
> We evaluated 5k image-text prompts using COCO 2017 validation set. While FID increases slightly, CLIP-Score remains the same. This indicates that the quality of images on other prompts remains comparable.
>
>
> | Model                | CLIPScore    | FID       |
> | -----                | ---------    | ----      |
> | SDXL                 | 21.17        | **83.95** |
> | FT. Inter(SDXL)  Ours| **21.20**    | 90.56	  |
>
>
> Similarly, Fig.3 shows humans favor output after finetuning on relations, which also indicates that the quality of images remains comparable. This shows the benefit of using LoRA finetuning to avoid catastrophic forgetting.
>
> We see this work as a focused study on compositional generalization and action-based role-bias, rather than generic image quality improvement, achieving sota across various prompts and image quality.
>
>
> > [w3] ... the last column of table 2 (VQAScore), ... the gap between unmatched and matched cases is reduced, but match/unmatch scores also decreased. That is, bias is decreased, but the model is not necessarily more accurate.
>
>
> We thank the reviewer for this important observation. We clarify in Section 4 (Bias).
>
>
> It shows **model is more accurate at showing correct action, action direction, and roles -- the goal of this paper** (e.g., correctly generating "mouse chasing cat" rather than collapsing into "cat chasing mouse"). Annotators preferred FT Inter Model preferred on against baselines on average 73%, and it outperformed baselines across all criteria according to $\beta$ in Table 2.
>
>
> Also, in Table 2, **Match score is increased by 1.3 points compared to SDXL (i.e. baseline with the same backbone). The lower absolute score is mainly relative to DALL-E 3, a much stronger model.** Hence the quality of generation is increased compared to the baseline.
>
>
> **Absolute Match/Unmatch VQAScores are unreliable for evaluating action-based relations and role bias**. Our human evaluation confirms this: **while DALL-E 3 achieves higher match scores, our model is more preferred by humans by 73%**. VLM/MLLM-based metrics inherit similar web data biases, are object-centric, and overlook complex/abstract relations (e.g. see [Yuksekgonul 2023] and others). Hence, we mostly consider human evaluation (we take 6 annotations per example; human-human correlation =0.8) and $\beta$ (it shows the highest correlation with human annotations) as more reliable metrics for this task.
>
>
>
>
> > [w4] ... line 181 that "We conjecture that these intermediates separately reinforce uncommon role bindings (e.g., mouse as chaser, cat as target), enabling the model to reconstruct the full rare relation more faithfully during generation." This conjecture was not theoretically or empirically discussed and little insights were given to support this claim.
>
> We thank the reviewer for the feedback.
>
> Our evaluation is specifically designed to provide empirical evidence to investigate this hypothesis. We measure improvement as generated images aligning with intended rare compositions (e.g., mouse chasing cat) rather than more freq counterparts (e.g. cat chasing mouse). We fine-tune T2I model on active/passive intermediate and evaluate through multiple ways -- they all show significant improvement in rare composition (e.g., mouse chasing cat) generation; hence better rare role binding (i.e., agent=mouse and patient=cat):
>
>
>
> * (1) **Human evaluation confirms the hypothesis** (Fig 3): Annotators preferred FT Inter over baselines on average by 73% on the rare compositions. Qualitative results (Fig. 4) show our method generates intended rare roles (mouse chasing cat) while baselines default to common patterns (cat chasing mouse).
>
>
> * (2) **Quantitative validation on holistic and fine-grained evaluation (Table 2)**: Our $\beta$ metric shows 15.20-point reduction in directional bias (23.10→7.90), demonstrating that intermediate training generates intended rare roles(e.g., mouse chasing cat)  rather more clearly while Table 1 showed that T2I baselines default to the common case (cat chasing mouse). Role-specific evaluation shows bias reduction across all criteria: spatial (5.10), orientation (2.10), pose (0.40), and facial expression (5.80). Confirming same finding in a more fine-grained manner.
>
>
> * (3) **Table 3 ablation & Fig.5** shows training only on active or passive intermediates helps, with combined training achieving best results. Also, increasing active weights better reinforces uncommon active roles (mouse as chaser, horse as rider), directly supporting our conjecture that intermediates separately strengthen specific role assignments.
>
> To communicate these points more clearly, we will refine the text in the results section and the captions of relevant tables and figures to explicitly discuss these conclusions.
>
> We hope that we have addressed the existing concerns and are happy to discuss more about them or if any other concerns exist.

---

> > ### Author Response · Authors · 2025-08-05
> >
> > Dear Reviewer YW6c, thank you again for your time and thoughtful feedback. We hope the additional COCO results and our detailed response in our rebuttal address your concerns. We’d like to kindly follow up and ask whether the questions are addressed. If you have further questions, we would be happy to discuss.

---

> > ### Comment · Reviewer_YW6c · 2025-08-06
> >
> > Thanks for your detailed response. I think you addressed majority of my concerns and I will adjust my score.

---

> > > ### Author Response · Authors · 2025-08-08
> > >
> > > Thanks for your prompt response. We’re glad our additional results and explanations helped address your concerns!

---

### Note · Authors · 2025-08-14

**What this paper is**. We study role bias in action-based relations. We find: (1) prevalence of common compositions in data hurts generation of inverse rare compositions; (2) compositional generation methods (layout/LLM-based) are ineffective for mitigating role bias (unlike attribute binding and capturing spatial relations). We provide the first simple, practical solution that significantly reduces role bias and enables rare compositional generation (e.g., mouse chasing cat).

**Strengths noted by reviewers**. Paper is well-written (Tzxi, 9JWJ). **Problem is significant**, shows **critical and underexplored bias**, and latest models exhibit this bias (9JWJ, jpRP, Tzxi). Idea is interesting (YW6c), **novel and sound** (9JWJ). **Analysis is in-depth, comprehensive, provides valuable insights, and confirms the hypothesis** (9JWJ,jpRP). Benchmark is reasonable (9JWJ), metrics appropriate (jpRP), and ablation important (Tzxi). Method is simple and effective with better alignment (jpRP, 9JWJ, YW6c).

**Added in rebuttal/discussion:** Clarifications and new evidence:
1. **Quantitative analysis of freq/rare compositions (9JWJ,jpRP):** 2 analyses: LLM log-probs and Google search counts. Reviewers acknowledged questions were addressed.
2. **Image quality & other prompts (YW6c):** On COCO captions, CLIPScore was same and FID comparable. YW6c indicated they would adjust the score.
3. **Breadth across models (Tzxi):** Beyond SDXL (arch: diffusion), we added SD3 (arch: rectified-flow transformer) in multi-relation and single-relation settings. Similar bias reductions were observed, indicating effectiveness across **distinct architectures**.

We're glad the rebuttal addressed all/most questions. Reviewers 9JWJ and jpRP remain positive. YW6c noted concerns were addressed, and adjusted their score. The only remaining point was "...multiple text-to-image models (...variations in architecture…)" by Tzxi. New results provide adequate evidence that active/passive intermediates generalize across distinct architectures (SDXL, SD3). This work offers novel insight into role bias and intermediate effectiveness; it doesn't propose novel models or alter architectures. SDXL was chosen as one of the most popular diffusions and most common in compositional methods for fair comparison.

Hence, we believe no major concerns remain after rebuttal. Paper can be updated by clarifying text, fixing typos, and adding rebuttal evidence without requiring significant additional experiments.

---

### Decision · Program_Chairs · 2025-09-17

**Decision:**

Accept (poster)

**Comment:**

This paper studied a quite interesting problem: the role binding in compositional image generation. The submission proposed a benchmark to evaluate how the order of subject/object affect the image generation model, and shows that current image generation model often fails to generate images for certain order of object and subjects (like "mouse chasing cat"), when the reverse order is popular. They also proposed a re-bind method with synthetic data generation in which they decompose triplets of such relations into intermediate triplets, and demonstrated fine-tuning with the synthetic data can mitigate the role binding bias.

Even though some reviewers pointed out some limitations like benmark data is not big, some manual work is needed to create the data, etc, however, after rebuttal, all reviewers gave positive scores and comments for the paper, for its interesting direction, novel data sets and methods, significant and practical impact to improve SOTA generative models, etc.

So I would recommend "Accept (poster)".